# Distinct contributions of functional and deep neural network features to representational similarity of scenes in human brain and behavior

Iris IA Groen[1,2]*, Michelle R Greene[3], Christopher Baldassano[4], Li Fei-Fei[5], Diane M Beck[6,7], Chris I Baker[1]

[1]Laboratory of Brain and Cognition, National Institutes of Health, Bethesda, United States; [2]Department of Psychology, New York University, New York City, United States; [3]Neuroscience Program, Bates College, Maine, United States; [4]Princeton Neuroscience Institute, Princeton University, Princeton, United States; [5]Stanford Vision Lab, Stanford University, Stanford, United States; [6]Department of Psychology, University of Illinois, Urbana-Champaign, United States; [7]Beckman Institute, University of Illinois, Urbana-Champaign, United States

**Abstract** Inherent correlations between visual and semantic features in real-world scenes make it difficult to determine how different scene properties contribute to neural representations. Here, we assessed the contributions of multiple properties to scene representation by partitioning the variance explained in human behavioral and brain measurements by three feature models whose inter-correlations were minimized *a priori* through stimulus preselection. Behavioral assessments of scene similarity reflected unique contributions from a functional feature model indicating potential actions in scenes as well as high-level visual features from a deep neural network (DNN). In contrast, similarity of cortical responses in scene-selective areas was uniquely explained by mid- and high-level DNN features only, while an object label model did not contribute uniquely to either domain. The striking dissociation between functional and DNN features in their contribution to behavioral and brain representations of scenes indicates that scene-selective cortex represents only a subset of behaviorally relevant scene information.
DOI: https://doi.org/10.7554/eLife.32962.001

*For correspondence:
iris.groen@nyu.edu

Competing interests: The authors declare that no competing interests exist.

## Introduction

Although researchers of visual perception often use simplified, highly controlled images in order to isolate the underlying neural processes, real-life visual perception requires the continuous processing of complex visual environments to support a variety of behavioral goals, including recognition, navigation and action planning (*Malcolm et al., 2016*). In the human brain, the perception of complex scenes is characterized by the activation of three scene-selective regions, the Parahippocampal Place Area (PPA; *Aguirre et al., 1998*; *Epstein and Kanwisher, 1998*), Occipital Place Area (OPA; *Hasson et al., 2002*; *Dilks et al., 2013*), and Medial Place Area (MPA; *Silson et al., 2016*), also referred to as the Retrosplenial Complex (*Bar and Aminoff, 2003*). A growing functional magnetic resonance imaging (fMRI) literature focuses on how these regions facilitate scene understanding by investigating what information drives neural responses in these regions when human observers view scene stimuli. Currently, a large set of candidate low- and high-level characteristics have been identified, including but not limited to: a scene's constituent objects and their co-occurrences; spatial layout; surface textures; contrast and spatial frequency, as well as scene semantics, contextual

associations, and navigational affordances (see *Epstein, 2014*; *Malcolm et al., 2016*; *Groen et al., 2017* for recent reviews).

This list of candidate characteristics highlights two major challenges in uncovering neural representations of complex real-world scenes (*Malcolm et al., 2016*). First, the presence of multiple candidate models merits careful comparison of the contribution of each type of information to scene representation. However, given the large number of possible models and the limited number that can realistically be tested in a single study, how do we select which models to focus on? Second, there are many inherent correlations between different scene properties. For example, forests are characterized by the presence of spatial boundaries and numerous vertical edges, whereas beaches are typically open with a prominent horizon, resulting in correlations between semantic category, layout and spatial frequency (*Oliva and Torralba, 2001*; *Torralba and Oliva, 2003*). This makes it problematic to explain neural representations of scenes based on just one of these properties (*Walther et al., 2009*; *Kravitz et al., 2011*; *Park et al., 2011*; *Rajimehr et al., 2011*) without taking into account their covariation. Indeed, an explicit test of spatial frequency, subjective distance and semantic properties found that due to inherent feature correlations, all three properties explained the same variance in fMRI responses to real-world scenes (*Lescroart et al., 2015*).

In the current fMRI study, we addressed the first challenge by choosing models based on a prior study that investigated scene categorization behavior (*Greene et al., 2016*). This study assessed the relative contributions of different factors that have traditionally been considered important for scene understanding, including a scene's component objects (e.g., *Biederman, 1987*) and its global layout (e.g, *Oliva and Torralba, 2001*), but also included novel visual feature models based on state-of-the-art computer vision algorithms (e.g., *Sermanet et al., 2013*) as well as models that reflect conceptual scene properties, such as superordinate categories, or the types of actions afforded by scene. Using an online same-different categorization paradigm on hundreds of scene categories from the SUN database (*Xiao et al., 2014*), a large-scale scene category distance matrix was obtained (reflecting a total of 5 million trials), which was subsequently compared to predicted category distances for the various candidate models. The three models that contributed most to human scene categorization were (1) a model based on human-assigned labels of actions that can be carried out in the scene ('functional model'), (2) a deep convolutional neural network ('DNN model') that was trained to map visual features natural images to a set of a 1000 image classes from the ImageNet database (*Deng et al., 2009*), and (3) human-assigned labels for all the objects in the scene ('object model'). Given the superior performance of these top three models in explaining scene categorization, we deemed these models most relevant to test in terms of their contribution to brain representations. Specifically, we determined the relative contribution of these three models to scene representation by comparing them against multi-voxel patterns in fMRI data collected while participants viewed a reduced set of scene stimuli from *Greene et al. (2016)*.

To address the second challenge, we implemented a stimulus selection procedure that reduced inherent correlations between the three models of interest *a priori*. Specifically, we compared predicted category distances for repeated samples of stimuli from the SUN database, and selected a final set of stimuli for fMRI for which the predictions were minimally correlated across the function, DNN and object model. To assess whether scene categorization behavior for this reduced stimulus set was consistent with the previous behavioral findings, participants additionally performed a behavioral multi-arrangement task outside the scanner. To isolate the unique contribution of each model to fMRI and behavioral scene similarity, we applied a variance partitioning analysis, accounting for any residual overlap in representational structure between models.

To anticipate, our data reveal a striking dissociation between the feature model that best describes behavioral scene similarity and the model that best explains similarity of fMRI responses in scene-selective cortex. While we confirmed that behavioral scene categorization was best explained a combination of the function model and DNN features, there was no unique representation of scene functions in scene-selective brain regions, which instead were best described by DNN features only. Follow-up analyses indicated that scene functions correlated with responses in regions outside of scene-selective cortex, some of which have been previously associated with action observation. However, a direct comparison between behavioral scene similarity and fMRI responses indicated that behavioral scene categorization correlated most strongly with scene-selective regions, with no discernible contribution of other regions. This dissociation between the features that contribute

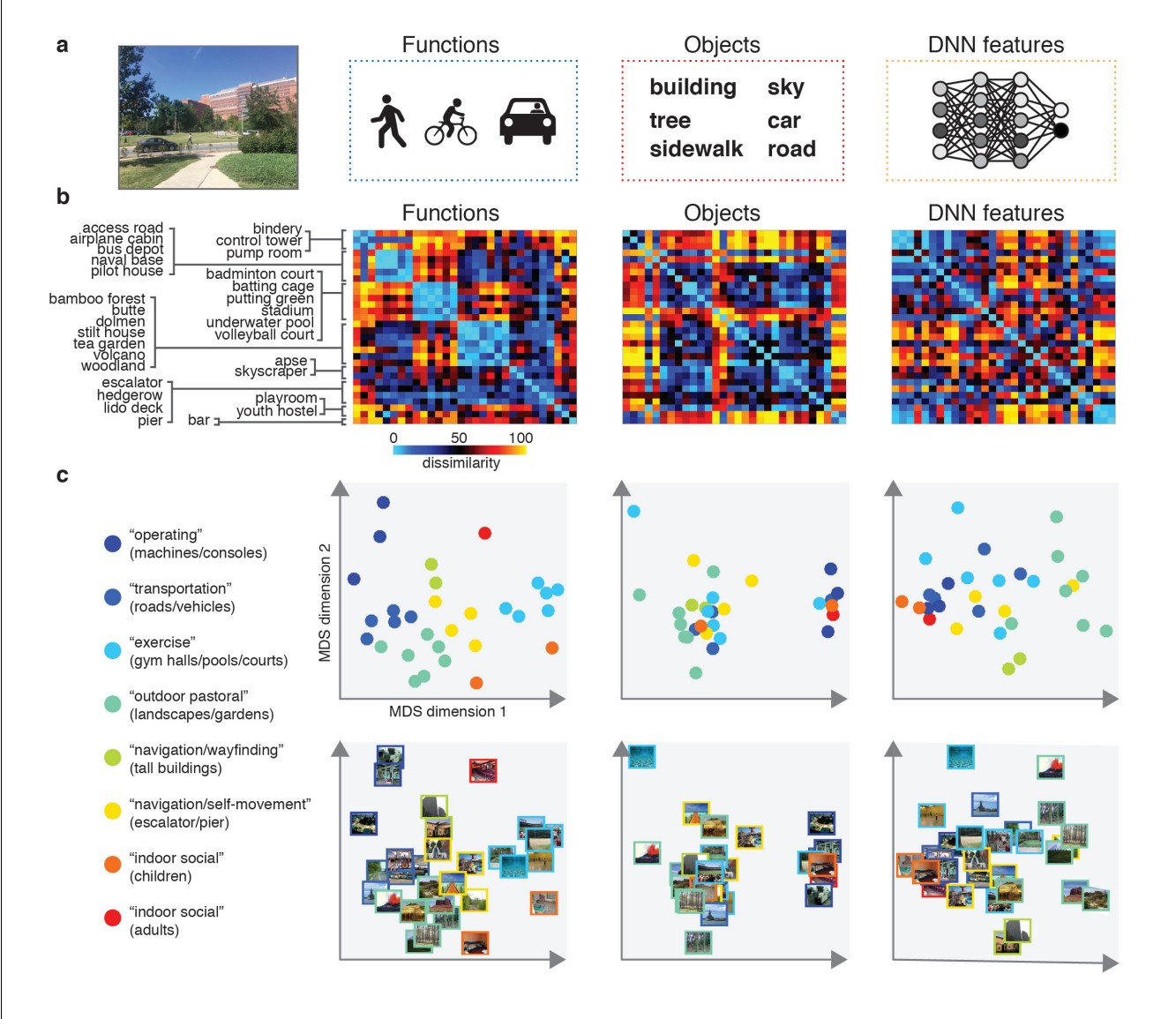

**Figure 1.** Models and predicted stimulus dissimilarity. (**A**) Stimuli were characterized in three different ways: functions (derived using human-generated action labels), objects (derived using human-generated object labels) and DNN features (derived using layer 7 of a 1000-class trained convolutional neural network). (**B**) RDMs showing predicted representational dissimilarity in terms of functions, objects and DNN features for the 30 scene categories sampled from *Greene et al. (2016)*. Scenes were sampled to achieve minimal between-RDM correlations. The category order in the RDMs is determined based on a k-means clustering on the functional RDM; clustering was performed by requesting eight clusters, which explained 80% of the variance in that RDM. RDMs were rank-ordered for visualization purposes only. (**C**) Multi-dimensional scaling plots of the model RDMs, color-coded based on the functional clusters depicted in B). Functional model clusters reflected functions such as 'sports', and 'transportation'; note however that these semantic labels were derived post-hoc after clustering, and did not affect stimulus selection. Critically, representational dissimilarity based on the two other models (objects and DNN features) predicted different cluster patterns. All stimuli and model RDMs, along with the behavioral and fMRI measurements, are provided in *Figure 1—source data 1*.

DOI: https://doi.org/10.7554/eLife.32962.002

The following source data is available for figure 1:

**Source data 1.**

DOI: https://doi.org/10.7554/eLife.32962.003

uniquely to behavioral versus fMRI scene similarity suggests that scene-selective cortex and DNN feature models represent only a subset of the information relevant for scene categorization.

## Results

### Disentangling function, DNN and object features in scenes

The goal of the study was to determine the contributions of function, DNN and object feature models to neural representations in scene-selective cortex. To do this, we created a stimulus set by iteratively sampling from the large set of scenes previously characterized in terms of these three types of information by *Greene et al. (2016)*. The DNN feature model was derived using a high-level layer of an AlexNet (*Krizhevsky et al., 2017*; *Sermanet et al., 2013*) that was pre-trained using ImageNet class labels (*Deng et al., 2009*), while the object and function feature models were derived based on object and action labels assigned by human observers through Amazon Mechanical Turk (see Materials and methods for details). On each iteration, pairwise distances between a subset of pseudo-randomly sampled categories were determined for each of these feature models, resulting in three representational dissimilarity matrices (RDMs) reflecting the predicted category distances for either the function, DNN or object model (*Figure 1A*) for that sample. Constraining the set to include equal numbers of indoor, urban, and natural landscape environments, our strategy was inspired by the odds algorithm of *Bruss (2000)*, in that we rejected the first 10,000 solutions, selecting the next solution that had lower inter-feature correlations than had been observed thus far. Thus, a final selection of 30 scene categories was selected in which the three RDMs were minimally correlated (Pearson's *r*: 0.23–0.26; *Figure 1B–C*; see Materials and methods).

Twenty participants viewed the selected scenes while being scanned on a high-field 7T Siemens MRI scanner using a protocol sensitive to blood oxygenation level dependent (BOLD) contrast (see Materials and methods). Stimuli were presented for 500 ms each while participants performed an orthogonal task on the fixation cross. To assess how each feature model contributed to scene categorization behavior for our much reduced stimulus set (30 instead of the 311 categories of *Greene et al., 2016*), participants additionally performed a behavioral multi-arrangement task (*Kriegeskorte and Mur, 2012*) on the same stimuli, administered on a separate day after scanning. In this task, participants were presented with all stimuli in the set arranged around a large white circle on a computer screen, and were instructed to drag-and-drop these scenes within the white circle according to their perceived similarity (see Materials and methods and *Figure 2A*).

### Function and DNN model both contribute uniquely to scene categorization behavior

To determine what information contributed to behavioral similarity judgments in the multi-arrangement task, we created RDMs based on each participant's final arrangement by measuring the pairwise distances between all 30 categories in the set (*Figure 2B*), and then computed correlations of these RDMs with the three model RDMs that quantified the similarity of the scenes in terms of either functions, objects, or DNN features, respectively (see *Figure 1B*).

Replicating *Greene et al. (2016)*, this analysis indicated that all three feature models were significantly correlated with scene categorization behavior, with the function model having the highest correlation on average (*Figure 2C*; objects: mean *r* = 0.16; DNN features: mean *r* = 0.26; functions: mean *r* = 0.29, Wilcoxon one-sided signed-rank test, all W(20) > 210, all *z* > 3.9, all p<0.0001). The correlation with functions was higher than with objects (Wilcoxon two-sided signed-rank test, W(20) = 199, *z* = 3.5, p=0.0004), but not than with DNN features (W(20) = 134, *z* = 1.1, p=0.28), which also correlated higher than objects (W(20) = 194, *z* = 3.3, p=0.0009). However, comparison at the level of individual participants indicated that functions outperformed both the DNN and object models for the majority of participants (highest correlation with functions: n = 12; with DNN features: n = 7; with objects: n = 1; *Figure 2D*).

While these correlations indicate that scene dissimilarity based on the function model best matched the stimulus arrangements that participants made, they do not reveal to what extent functional, DNN or object features *independently* contribute to the behavior. To assess this, we performed two additional analyses. First, we computed partial correlations between models and behavior whereby the correlation of each model with the behavior was determined whilst taking into account the contributions of the other two models. The results indicated that each model independently contributed to the behavioral data: significant partial correlations were obtained for the object (W(20) = 173, *z* = 2.5, p=0.006), DNN features (W(20) = 209, *z* = 3.9, p<0.0001) and function

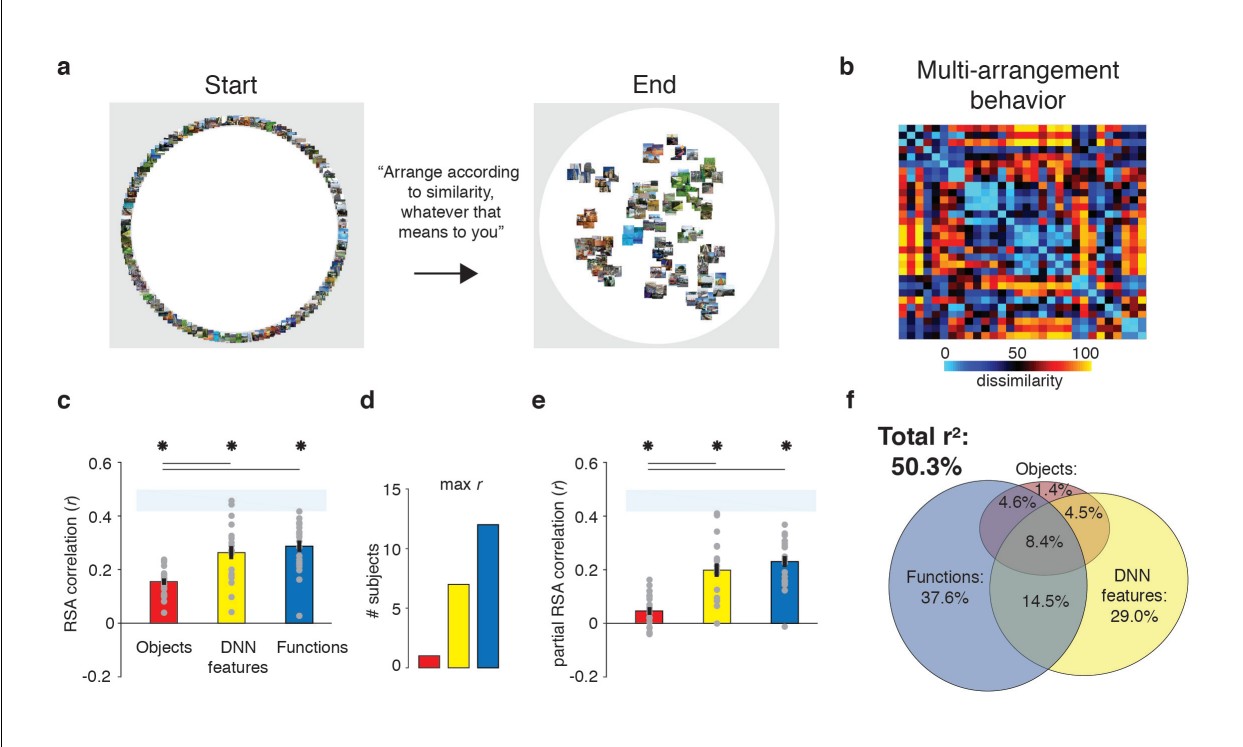

**Figure 2.** Behavioral multi-arrangement paradigm and results. (A) Participants organized the scenes inside a large white circle according to their perceived similarity as determined by their own judgment, without receiving instructions as to what information to use to determine scene similarity. (B) RDM displaying the average dissimilarity between categories in the behavioral arrangements, ordered the same way as *Figure 1B* (rank-ordered for visualization only). (C) Average (bar) and individual participant (gray dots) correlations between the behavioral RDM and the model RDMs for objects (red), DNN features (yellow) and functions (blue). Stars (*) indicate p<0.05 for model-specific one-sided signed-rank tests against zero, while horizontal bars indicate p<0.05 for two-sided pairwise signed-rank tests between models; *p*-values were FDR-corrected across both types of comparisons. The light-blue shaded rectangular region reflects the upper and lower bound of the noise ceiling, indicating RDM similarity between individual participants and the group average (see Materials and methods). Error bars reflect SEM across participants. (D) Count of participants whose behavioral RDM correlated highest with either objects, DNN features or functions. (E) Partial correlations for each model RDM. Statistical significance was determined the same way as in C. (F) Euler diagram depicting the results of a variance partitioning analysis on the behavioral RDM for objects (red circle), DNN features (yellow circle) and functions (blue circle). Unique (non-overlapping diagram portions) and shared (overlapping diagram portions) variances are expressed as percentages of the total variance explained by all models combined.
DOI: https://doi.org/10.7554/eLife.32962.004

models (W(20) = 209, $z$ = 3.9, p<0.0001), with the function model having the largest partial correlation (*Figure 2E*). Direct comparisons yielded a similar pattern as the independent correlations, with weaker contributions of objects relative to both functional (W(20) = 201, $z$ = 3.6, p<0.0003) and DNN features (W(20) = 195, $z$ = 3.4, p=0.0008), whose partial correlations did not differ from one another (W(20) = 135, $z$ = 1.12, p=0.26).

Second, we conducted a variance partitioning analysis, in which the function, DNN and object feature models were entered either separately or in combination as predictors in a set of multiple regression analyses aimed at explaining the multi-arrangement categorization behavior. By comparing the explained variance based on regression on individual models versus models in combination, we computed portions of unique variance contributed by each model as well as portions of shared variance across models (see Materials and methods for details).

A full model in which all three models were included explained 50.3% of the variance in the average multi-arrangement behavior (*Figure 2F*). Highlighting the importance of functional features for scene categorization, the largest portion of this variance could be uniquely attributed to the function model (unique $r^2$ = 37.6%), more than the unique variance explained by the DNN features (unique $r^2$ = 29.0%) or the objects (unique $r^2$ = 1.4%). This result is consistent with the findings of *Greene et al. (2016)*, who found unique contributions of 45.2% by the function model, 7.1% by the

DNN model and 0.3% by objects, respectively. (When performing the variance partitioning on the behavioral categorization measured in *Greene et al. (2016)* but limited to the reduced set of 30 scene categories used here, we obtained a highly similar distribution of unique variances as for the current behavioral data, namely 42.8% for functions, 28.0% for DNN features, and 0.003% for objects, respectively. This suggests that the higher contribution of the DNN relative to *Greene et al. (2016)* is a result of the reduced stimulus set used here, rather than a qualitative difference in experimental results between studies.) One interesting difference with this previous study is that the degree of shared variance between all three models is notably smaller (8.4% versus 27.4%). This is presumably a result of our stimulus selection procedure that was explicitly aimed at minimizing correlations between models. Importantly, a reproducibility test indicated that the scene similarity reflected in the multi-arrangement behavior was highly generalizable, resulting in an RDM correlation of $r = 0.73$ (95% confidence interval = [0.73–0.88], p=0.0001) across two different sets of scene exemplars that were evenly distributed across participants (see Materials and methods).

In sum, these results confirm a unique, independent contribution of the function model to scene categorization behavior, here assessed using a multi-arrangement sorting task (as opposed to a same/different categorization task). We also found a unique but smaller contribution of the DNN feature model, while the unique contribution of the object model was negligible. Next, we examined to what extent this information is represented in brain responses to the same set of scenes as measured with fMRI.

## DNN model uniquely predicts responses in scene-selective cortex

To determine the information that is represented in scene-selective brain regions PPA, OPA and MPA, we created RDMs based on the pairwise comparisons of multi-voxel activity patterns for each scene category in these cortical regions (*Figure 3A*), which we subsequently correlated with the RDMs based on the object, function and DNN feature models. Similar to the behavioral findings, all three feature models correlated with the fMRI response patterns in PPA (objects: $W(20) = 181$, $z = 2.8$, p=0.002; DNN: $W(20) = 206$, $z = 3.8$, p<0.0001; functions: $W(20) = 154$, $z = 1.8$, p=0.035, see *Figure 3B*). However, PPA correlated more strongly with the DNN feature model than the object ($W(20) = 195$, $z = 2.5$, p=0.012) and function ($W(20) = 198$, $z = 3.5$, p<0.0005) models, which did not differ from one another ($W(20) = 145$, $z = 1.5$, p=0.14). In OPA, only the DNN model correlated with the fMRI response patterns ($W(20) = 165$, $z = 2,2$, p=0.013), and this correlation was again stronger than for the object model ($W(20) = 172$, $z = 2.5$, p=0.012), but not the function model ($W(20) = 134$, $z = 1.1$, p=0.28). In MPA, no correlations were significant (all $W(14) < 76$, all $z < 1.4$, all p>0.07).

When the three models were considered in combination, only the DNN model yielded a significant partial correlation (PPA: $W(20) = 203$, $z = 3.6$, p<0.0001, OPA: $W(20) = 171$, $z = 2.5$, p=0.007, *Figure 3C*), further showing that DNN features best capture responses in scene-selective cortex. No significant partial correlation was found for the object model (PPA: $W(20) = 148$, $z = 1.6$, p=0.056; OPA: $W(20) = 74$, $z = 1.2$, p=0.88) or the function model (PPA: $W(20) = 98$, $z = 0.3$, p=0.61, OPA: $W(20) = 127$, $z = 0.8$, p=0.21), or for any model in MPA (all $W(14) < 63$, all $z < 0.66$, all p>0.50). Variance partitioning of the fMRI RDMs (*Figure 3D*) indicated that the DNN model also contributed the largest portion of unique variance: in PPA and OPA, DNN features contributed 71.1% and 68.9%, respectively, of the variance explained by all models combined, more than the unique variance explained by the object (PPA: 5.3%; OPA, 2.3%) and function (PPA: 0.3%; OPA: 2.6%) models. In MPA, a larger share of unique variance was found for the function model (41.5%) than for the DNN (38.7%) and object model (3.2%); however, overall explained variance in MPA was much lower than in the other ROIs. A reproducibility test indicated that RDMs generalized across participants and stimulus sets for PPA ($r = 0.26$ [0.03–0.54], p=0.009) and OPA ($r = 0.23$ [0.04–0.51], p=0.0148), but not in MPA ($r = 0.06$ [−0.16–0.26], p=0.29), suggesting that the multi-voxel patterns measured in MPA were less stable (this is also reflected in the low noise ceiling in MPA in *Figure 3B and C*).

Taken together, the fMRI results indicate that of the three models considered, deep network features (derived using a pre-trained convolutional network) best explained the coding of scene information in PPA and OPA, more so than object or functional information derived from semantic labels that were explicitly generated by human observers. For MPA, results were inconclusive, as none of the models adequately captured the response patterns measured in this region, which also did not generalize across stimulus sets and participants. This result reveals a discrepancy between measurements of brain responses versus behavioral scene similarity, which indicated a large contribution of

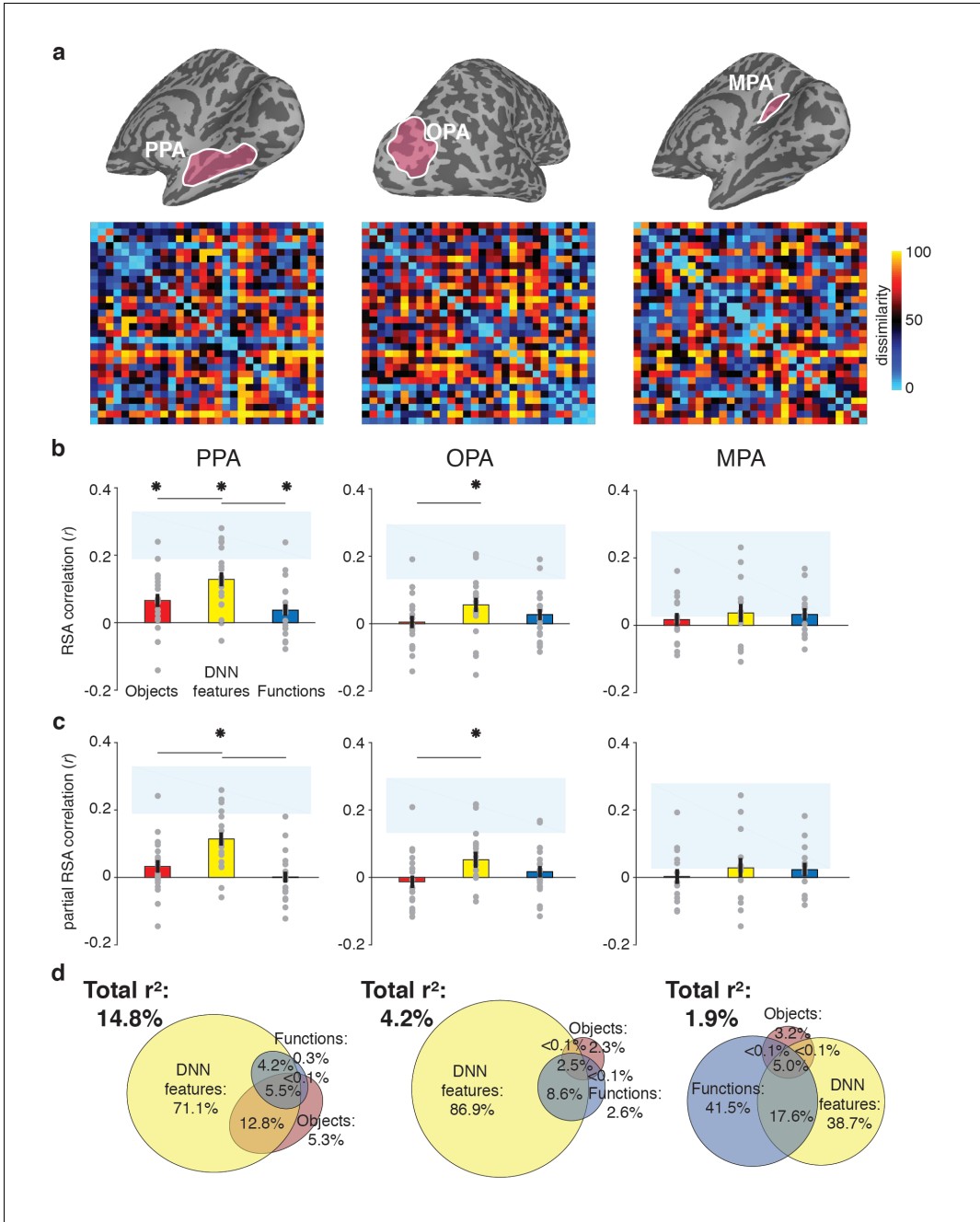

**Figure 3.** RDMs and model comparisons for fMRI Experiment 1 (n = 20). (A) RDMs displaying average dissimilarity between categories in multi-voxel patterns in PPA, OPA and MPA, ordered as in *Figure 1B* (rank-ordered for visualization only). (B) Average (bar) and individual participant (gray dots) correlations between the ROIs in A) and the model RDMs for objects (red), DNN features (yellow) and functions (blue) (FDR-corrected). See legend of *Figure 2B* for explanation of the statistical indicators and noise ceiling. (C) Partial correlations for each model RDM. Statistics are the same as in B). (D) Euler diagram depicting results of variance partitioning the average dissimilarity in each ROI between models, expressed as percentages of unique and shared variance of the variance explained by all three models together.

DOI: https://doi.org/10.7554/eLife.32962.005

functions to scene representation independent of the DNN features. To better understand if and how scene-selective cortex represents behaviorally relevant information, we next compared measurements of behavioral scene similarity to the fMRI responses directly.

## Scene-selective cortex correlation with behavior reflects DNN model

To assess the extent to which fMRI response patterns in scene-selective cortex predicted behavioral scene similarity, we correlated each of the scene-selective ROIs with three measures of behavioral categorization: (1) the large-scale online categorization behavior measured in *Greene et al. (2016)*, (2) the average multi-arrangement behavior of the participants in the current study, and (3) each individual participant's own multi-arrangement behavior. This analysis revealed a significant correlation with behavior in all scene-selective ROIs (*Figure 4A*). In PPA, all three measures of behavioral categorization correlated with fMRI responses (signed-rank test, online categorization behavior: W(20) = 168, z = 2.3, p=0.010; average multi-arrangement behavior: W(20) = 195, z = 3.3, p=0.0004; own arrangement behavior: W(20) = 159, z = 2.0, p=0.023). In OPA, significant correlations were found for both of the average behavioral measures (online categorization behavior: W(20) = 181, z = 2.8, p=0.002; average multi-arrangement behavior: W(20) = 158, z = 1.96, p=0.025), but not for the participant's own multi-arrangement behavior (W(20) = 106, z = 0.02, p=0.49), possibly due to higher noise in the individual data. Interestingly, however, MPA showed the opposite pattern: participant's own behavior was significantly related to the observed fMRI responses (W(14) = 89, z = 2.26, p=0.011), but the average behavioral measures were not (online behavior: W(14) = 47, z = 0.4, p=0.65; average behavior: W(14) = 74, z = 1.3, p=0.09). Combined with the reproducibility test results (see above), this suggests that the MPA responses are more idiosyncratic to individual participants or stimulus sets.

While these results support an important role for scene-selective regions in representing information that informs scene categorization behavior, they also raise an intriguing question: what aspect of categorization behavior is reflected in these neural response patterns? To address this, we performed another variance partitioning analysis, now including the average multi-arrangement behavior as a predictor of the fMRI response patterns, in combination with the two models that correlated most strongly with this behavior, that is the DNN and function models. The purpose of this analysis was to determine how much variance in neural responses each of the models *shared* with the behavior, and whether there was any behavioral variance in scene cortex that was not explained by our models. If the behaviorally relevant information in the fMRI responses is primarily of a functional nature, we would expect portions of the variance explained by behavior to be shared with the function model. Alternatively, if this variance reflects mainly DNN features (which also contributed uniquely to the behavioral categorization; *Figure 2F*), we would expect it to be shared primarily with the DNN model.

Consistent with this second hypothesis, the variance partitioning results indicated that in OPA and PPA, most of the behavioral variance in the fMRI response patterns was shared with the DNN model (*Figure 4B*). In PPA, the behavioral RDMs on average shared 25.7% variance with the DNN model, while a negligible portion was shared with the function model (less than 1%); indeed, nearly all variance shared between the function model and the behavior was also shared with the DNN model (10.1%). In OPA, a similar trend was observed, with behavior sharing 38.9% of the fMRI variance with the DNN model. In OPA, the DNN model also eclipsed nearly all variance that behavior shared with the function model (9.7% shared by behavior, functions and DNN features), leaving only 1.6% of variance shared exclusively by functions and behavior. In contrast, in MPA, behavioral variance was shared with either the DNN model or the function model to a similar degree (14.7% and 17.7%, respectively), with an additional 27.1% shared with both; note, however, again MPA's low explained variance overall.

In sum, while fMRI response patterns in PPA and OPA reflect information that contributes to scene similarity judgments, this information aligns best with the DNN feature model; it does not reflect the unique contribution of functions to scene categorization behavior. While in MPA, the behaviorally relevant representations may partly reflect other information, the overall explained variance in MPA was again quite low, limiting interpretation of this result.

## Relative model contributions to fMRI responses do not change with task manipulation

An important difference between the behavioral and the fMRI experiment was that participants had access to the entire stimulus set when performing the behavioral multi-arrangement task, which they could perform at their own pace, while they performed a task unrelated to scene categorization in

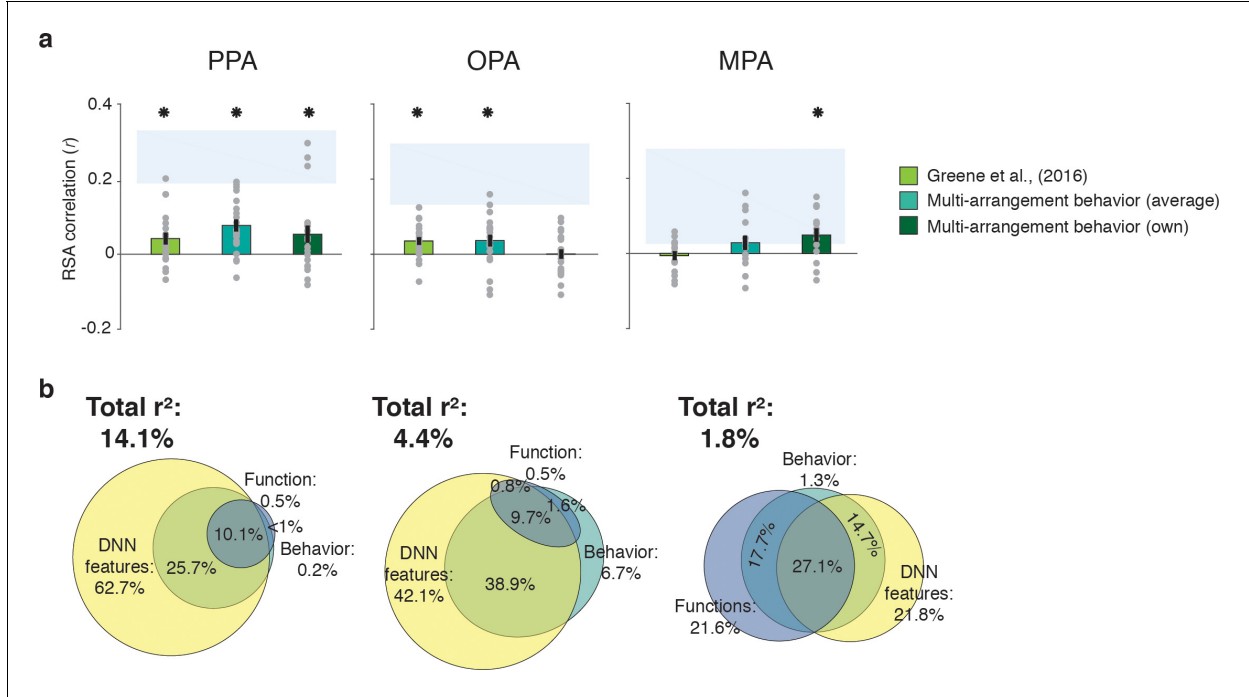

**Figure 4.** Correlations and variance partitioning of behavioral measurements of scene categorization and similarity of fMRI responses. (**A**) Correlations of three measures of behavioral categorization (see Results section for details) with fMRI response patterns in PPA, OPA and MPA. See legend of *Figure 2B* for explanation of the statistical indicators and noise ceiling. (**B**) Euler diagram depicting the results of variance partitioning the fMRI responses in PPA, OPA and MPA for DNN features (yellow), functions (blue) and average sorting behavior (green), indicating that the majority of the variance in the fMRI signal that is explained by categorization behavior is shared with the DNN features.

DOI: https://doi.org/10.7554/eLife.32962.006

the fMRI scanner. Therefore, we reasoned that a possible explanation of the discrepancy between the fMRI and behavioral findings could be a limited engagement of participants with the briefly presented scenes while in the scanner, resulting in only superficial encoding of the images in terms of visual features that are well captured by the DNN model, rather than functional or object features that might be more conceptual in nature.

To test this possible explanation, we ran Experiment 2 and collected another set of fMRI data using the exact same visual stimulation, but with a different task instruction (n = 8; four of these participants also participated in Experiment 1, allowing for direct comparison of tasks within individuals). Specifically, instead of performing an unrelated fixation task, participants covertly named the presented scene. Covert naming has been shown to facilitate stimulus processing within category-selective regions and to enhance semantic processing (*van Turennout et al., 2000*; *van Turennout et al., 2003*). Before entering the scanner, participants were familiarized with all the individual scenes in the set, whereby they explicitly generated a name for each individual scene (see Materials and methods). Together, these manipulations were intended to ensure that participants attended to the scenes and processed their content to a fuller extent than in Experiment 1.

Despite this task difference, Experiment 2 yielded similar results as Experiment 1 (*Figure 5A*). Reflecting participant's enhanced engagement with the scenes when performing the covert naming task, overall model correlations were considerably higher than in Experiment 1, and now yielded significant correlations with the function model in both OPA and MPA (*Figure 5B*). The direct test of reproducibility also yielded significant, and somewhat increased, correlations for PPA (r = 0.35 [0.26–0.55], p=0.0001) and OPA (r = 0.27 [0.18–0.60], p=0.039), but not in MPA (r = 0.10 [−0.07–0.28], p=0.17). Importantly, in all three ROIs, the DNN model correlations were again significantly stronger than the function and object model correlations, which again contributed very little unique variance (*Figure 5C*). Direct comparison of RDM correlations across the two Experiments indicated that in PPA and OPA, the naming task resulted in increased correlations for the DNN model only

(two-sided Wilcoxon ranksum test, PPA: p=0.0048; OPA p=0.0056), without any difference in correlations for the other models (all p>0.52). In MPA, none of the model correlations differed across tasks (all p>0.21). Increased correlation with the DNN model was present within the participants that participated in both experiments (n = 4; see Materials and methods): in PPA and OPA, 4/4 and 3/4 participants showed an increased correlation, respectively, whereas no consistent patterns was observed for the other models and MPA (*Figure 5D*).

In sum, the results of Experiment 2 indicate that the strong contribution of DNN features to fMRI responses in scene-selective cortex is not likely the result of limited engagement of participants with the scenes when viewed in the scanner. If anything, enhanced attention to the scenes under an explicit naming instruction resulted in even stronger representation of the DNN features, without a clear increase in contributions of the function or object models.

## Contributions of the function model outside scene-selective cortex

Our results so far indicate a dissociation between brain and behavioral assessments of the representational similarity of scenes. In the behavioral domain, visual features in a deep convolutional network uniquely contributed to behavioral scene categorization, but the function model also exhibited a large unique contribution, regardless of whether this behavior was assessed using a same-different categorization or a multi-arrangement task. In contrast, fMRI responses in scene-selective cortex were primarily driven by DNN features, without convincing evidence of an independent contribution of functions. Given this lack of correlation with the function model in the scene-selective cortex, we explored whether this information could be reflected elsewhere in the brain by performing whole-brain searchlight analyses. Specifically, we extracted the multi-voxel patterns from spherical ROIs throughout each participant's entire brain volume and performed partial correlation analyses including all three models (DNN features, objects, functions) to extract corresponding correlation maps for each model. The resulting whole-brain searchlight maps were then fed into a to surface-based group analysis (see Materials and methods) to identify clusters of positive correlations indicating significant model contributions to brain representation throughout all measured regions of cortex.

The results of these searchlight analyses were entirely consistent with the ROI analyses: for the DNN feature model, significant searchlight clusters were found in PPA and OPA (*Figure 6A*), but not MPA, whereas no significant clusters were found for the function model in any of the scene-selective ROIs. (The object model yielded no positive clusters). However, two clusters were identified for the function model outside of scene-selective cortex (*Figure 6B*): 1) a bilateral cluster on the ventral surface, lateral to PPA, overlapping with the fusiform and temporal lateral gyri, and 2) a unilateral cluster on the left lateral surface, located adjacent to, but more ventral than, OPA, overlapping the posterior middle and inferior temporal gyrus.

The observed dissociation between behavioral categorization and scene-selective cortex suggests that the functional features of scenes that we found to be important for scene categorization behavior are potentially represented outside of scene-selective cortex. If so, we would expect the searchlight clusters that correlated with the function model to show a correspondence with the behavioral scene categorization. To test this, we directly correlated the multi-arrangement behavior with multi-voxel pattern responses throughout the brain. Consistent with the results reported in *Figure 4*, we found a significant searchlight correlation between the behavioral measurements and response patterns in PPA and OPA (*Figure 7A*). Surprisingly, however, behavioral categorization did not correlate with any regions outside these ROIs, including the clusters that correlated with the function model.

In order to better understand how representational dissimilarity in those clusters related to the functional feature model, we extracted the average RDM from each searchlight cluster and inspected which scene categories were grouped together in these ROIs. Visual inspection of the RDM and MDS plots of the RDMs (*Figure 7B*) indicates that in both the bilateral ventral and left-lateralized searchlight clusters, there is some grouping by category according to the function feature model (indicated by grouping by color in the MDS plot). However, it is also clear that the representational space in these ROIs does not *exactly* map onto the functional feature model in *Figure 1C*. Specifically, a few categories clearly 'stand out' with respect to the other categories, as indicated by a large average distance relative to the other categories in the stimulus set. Most of the scene categories that were strongly separated all contained scene exemplars depicting humans that performed actions (see *Figure 7C*), although it is worth noting that scene exemplars in the fourth most distinct category, 'volcano', did not contain any humans but may be characterized by implied motion. These

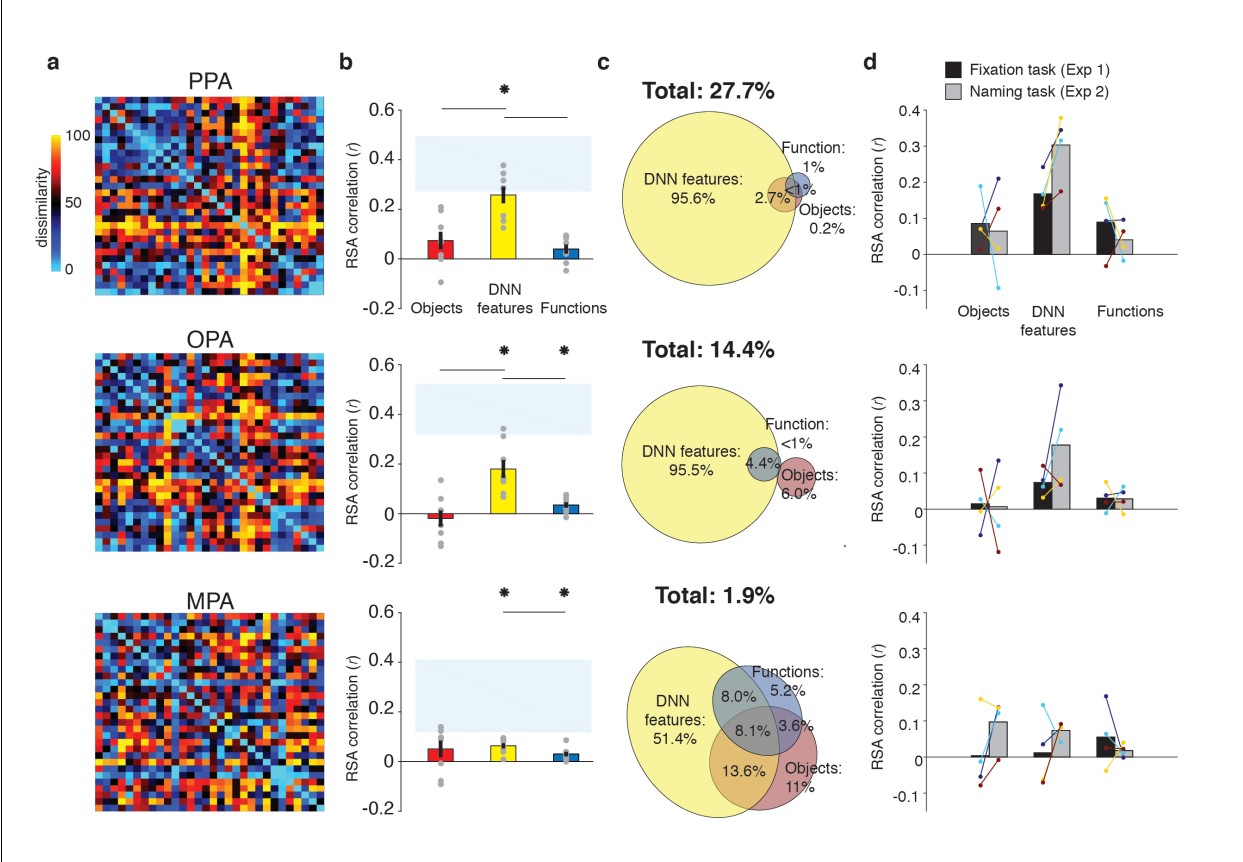

**Figure 5.** RDMs and model comparisons for Experiment 2 (n = 8, covert naming task). (**A**) Average dissimilarity between categories in multi-voxel patterns measured in PPA, OPA and MPA (rank-ordered as in **Figure 1B**). (**B**) Correlations between the ROIs in A) and the model RDMs for objects (red), DNN features (yellow) and functions (blue) (FDR-corrected). See legend of **Figure 2B** for explanation of the statistical indicators and noise ceiling. Note how in PPA, the DNN model correlation approaches the noise ceiling, suggesting that this model adequately captures the information reflected in this ROI. (**C**) Euler diagram depicting the results of variance partitioning the average dissimilarity in each ROI. (**D**) Average (bars) and individual (dots/lines) within-participant (n = 4) comparison of fMRI-model correlations across the different task manipulations in Experiment 1 and 2 (participants were presented with a different set of scenes in each task, see Materials and methods). Note how covert naming mainly enhances the correlation with DNN features.

DOI: https://doi.org/10.7554/eLife.32962.007

post-hoc observations suggest that (parts of) the searchlight correlation with the functional feature model may be due to the presence of human-, body- and/or motion selective voxels in these searchlight clusters.

In sum, the searchlight analyses indicate that the strongest contributions of the DNN model were found in scene-selective cortex. While some aspects of the function model may be reflected in regions outside of scene-selective cortex, these regions did not appear to contribute to the scene categorization behavior, and may reflect selectivity for only a subset of scene categories that clustered together in the function model.

## Scene-selective cortex correlates with features from both mid- and high-level DNN layers

Our results highlight a significant contribution of DNN features to representations in scene-selective cortex. DNNs consist of multiple layers that capture a series of transformations from pixels in the input image to a class label, implementing a non-linear mapping of local convolutional filters responses (layers 1–5) onto a set of fully-connected layers that consist of classification nodes (layers 6–8) culminating in a vector of output 'activations' for labels assigned in the DNN training phase. Visualization and quantification methods of the learned feature selectivity (e.g., *Zhou et al., 2014*;

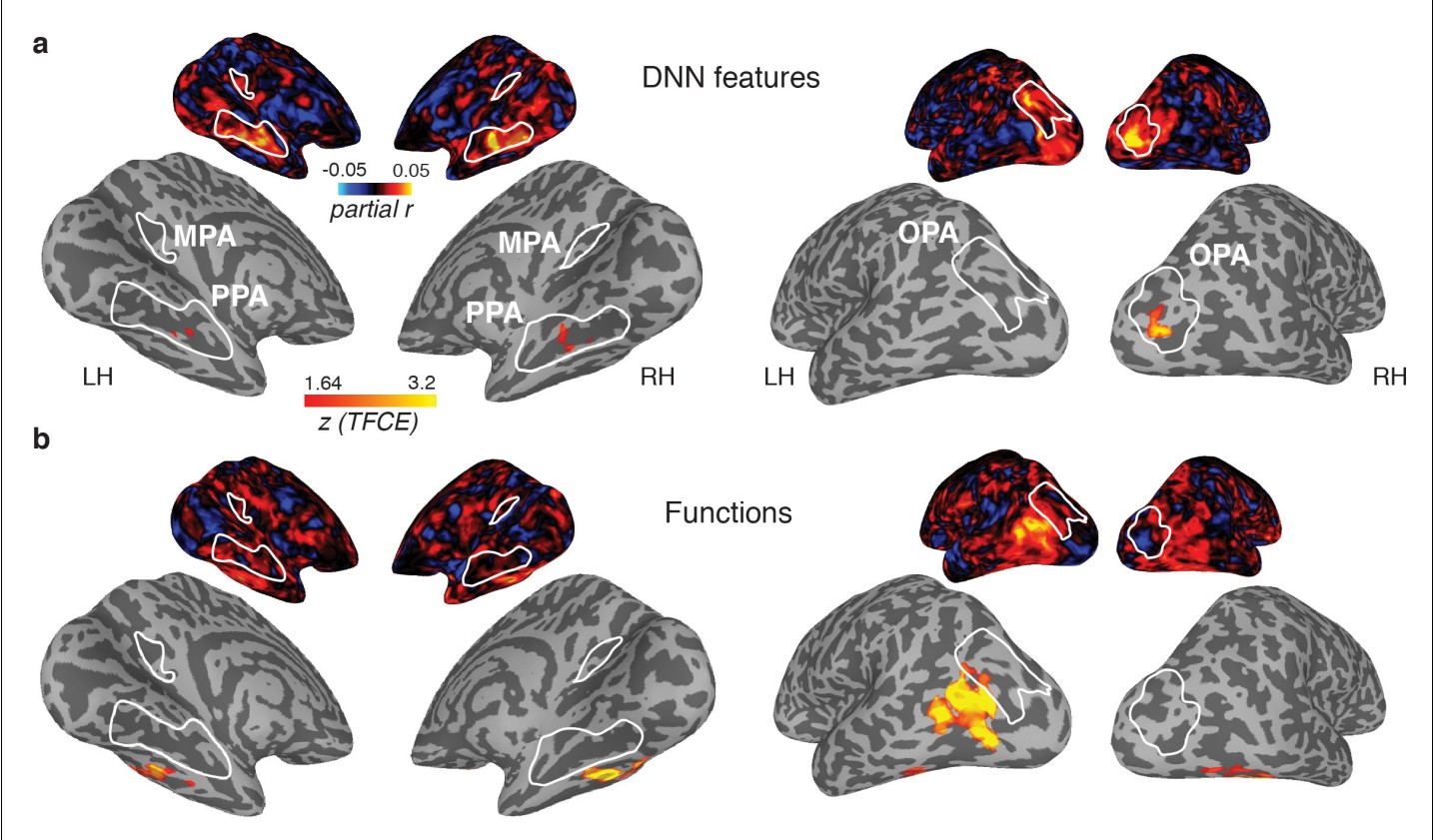

**Figure 6.** Medial (left) and lateral (right) views of group-level searchlights for (A) the DNN and (B) function model, overlaid on surface reconstructions of both hemispheres of one participant. Each map was created by submitting the partial correlation maps for each model and hemisphere to one-sample tests against a mean of zero, cluster-corrected for multiple comparisons using Threshold-Free Cluster Enhancement (thresholded on z = 1.64, corresponding to one-sided p<0.05). Unthresholded versions of the average partial correlation maps are inset above. Group-level ROIs PPA, OPA and MPA are highlighted in solid white lines. Consistent with the ROI analyses, the DNN feature model contributed uniquely to representation in PPA and OPA. The function model uniquely correlated with a bilateral ventral region, as well as a left-lateralized region overlapping with the middle temporal and occipital gyri.

DOI: https://doi.org/10.7554/eLife.32962.008

*Güçlü and van Gerven, 2015*; *Bau et al., 2017*; *Wen et al., 2017*) suggest that while earlier layers contain local filters that resemble V1-like receptive fields, higher layers develop selectivity for entire objects or object parts, perhaps resembling category-selective regions in visual cortex. Our deep network feature model was derived using a single high-level layer, fully-connected layer 7 ('fc7'). Moreover, this model was derived using the response patterns of a DNN that was pretrained on ImageNet (*Deng et al., 2009*), an image database largely consisting of object labels. Given the strong performance of the DNN feature model in explaining the fMRI responses in scene-selective cortex, it is important to determine whether this result was exclusive to higher DNN layers, and whether the task used for DNN training influences how well the features represented in individual layers explain responses in scene-selective cortex. To do so, we conducted a series of exploratory analyses to assess the contribution of other DNN layers to fMRI responses, whereby we compared DNNs that were trained using either object or scene labels.

To allow for a clean test of the influence of DNN training on features representations in each layer, we derived two new sets of RDMs by passing our stimuli through (1) a novel 1000-object label ImageNet-trained network implemented in Caffe (*Jia et al., 2014*) ('ReferenceNet') and (2) a 250-scene label Places-trained network ('Places') (*Zhou et al., 2014*), (see Materials and methods). Direct comparisons of the layer-by-layer RDMs of these two DNNs (*Figure 8A*) indicated that both models extracted similar features, evidenced by strong between-model correlations overall (all layers r > 0.6). However, the similarity between models decreased with higher layers, suggesting that

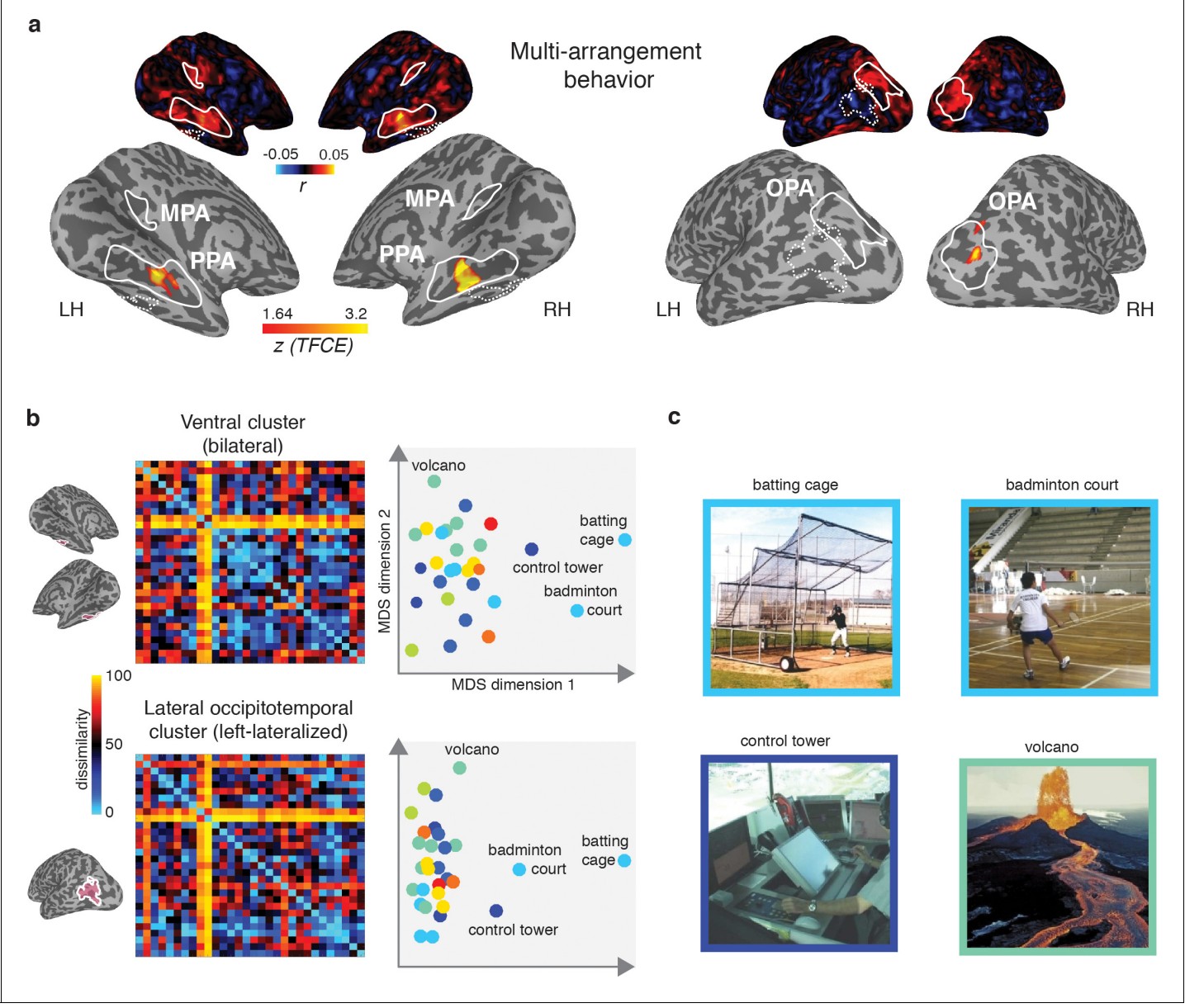

**Figure 7.** Multi-arrangement behavior searchlights and post-hoc analysis of functional clusters. (A) Searchlight result for behavioral scene categorization. Maps reflect correlation (Pearson's *r*) of the group-average behavior in the multi-arrangement task from the participants of Experiment 1. Scene-selective ROIs are outlined in white solid lines; the searchlight clusters showing a significant contribution of the functional model are outlined in dashed white lines for reference. See *Figure 6* for further explanation of the searchlight display. (B) RDM and MDS plots based on the MVPA patterns in the function model searchlight clusters. RDM rows are ordered as in *Figure 1B* and category color coding in the MDS plots is as in *Figure 1C*. (C) Illustrative exemplars of the four categories that were most dissimilar from other categories within the searchlight-derived clusters depicted in B.

DOI: https://doi.org/10.7554/eLife.32962.009

features in higher DNN layers become tailored to the task they are trained on. Moreover, this suggests that higher layers of the scene-trained DNN could potentially capture different features than the object-trained DNN. To investigate this, we next computed correlations between the features in each DNN layer and the three original feature models (*Figure 8B*).

As expected, the original fc7 DNN model (which was derived using DNN responses to the large set of images in the *Greene et al. (2016)* database, and thus not corresponding directly to the reduced set of stimuli used in the current study) correlated most strongly with the new DNN layer

representations, showing steadily increasing correlations with higher layers of both object-trained and the scene-trained DNN. By design, the object and functional feature models should correlate minimally with layer 7 of the object-trained ReferenceNet DNN. However, the function model correlated somewhat better with higher layers of the scene-trained DNN, highlighting a potential overlap of the function model with the scene-trained DNN features, again suggesting that the higher layers of the scene-trained DNN potentially capture additional information that is not represented in the object-trained DNN. Therefore, we next tested whether the scene-trained DNN correlated more strongly with fMRI responses in scene-selective cortex.

Layer-by-layer correlations of the object-trained (*Figure 8C*) and the scene-trained DNN (*Figure 8D*) with fMRI responses in PPA, OPA and MPA however did not indicate a strong difference in DNN performance as a result of training. In PPA, both the object-trained and place-trained DNN showed increased correlation with higher DNN layers, consistent with previous work showing a hierarchical mapping of DNN layers to low vs. high-level visual cortex (*Güçlü and van Gerven, 2015*; *Cichy et al., 2016*; *Wen et al., 2017*). Note however that the slope of this increase is quite modest; while higher layers overall correlate better than layers 1 and 2, in both DNNs the correlation with layer three is not significantly different from the correlation of layers 7 and 8. In OPA, we observed no evidence for increased performance with higher layers for the object-trained DNN; none of the pairwise tests survived multiple comparisons correction. In fact, for the scene-trained DNN, the OPA correlation significantly *decreased* rather than increased with higher layers, showing a peak correlation with layer 3. No significant correlations were found for any model layer with MPA. These observations were confirmed by searchlight analyses in which whole-brain correlation maps were derived for each layer of the object- and scene-trained DNN: see *Figure 8—video 1* and *Figure 8—video 2* for layer-by-layer searchlight results for the ReferenceNet and the Places DNN, respectively.

These results indicate that despite a divergence in representation in high-level layers for differently-trained DNNs, their performance in predicting brain responses in scene-selective cortex is quite similar. In PPA, higher layers perform significantly better than (very) low-level layers, but mid-level layers already provide a relatively good correspondence with PPA activity. This result was even more pronounced for OPA where mid-level layers yielded the maximal correlations for both DNNs regardless of training. Therefore, these results suggest that fMRI responses in scene-selective ROIs may reflect a contribution of visual DNN features of intermediate complexity rather than, or in addition to, the fc7 layer that was selected *a priori*.

## Discussion

We assessed the contribution of three scene feature models previously implicated to be important for behavioral scene understanding to neural representations of scenes in the human brain. First, we confirmed earlier reports that functions strongly contribute to scene categorization by replicating the results of *Greene et al. (2016)*, now using a multi-arrangement task. Second, we found that brain responses to visual scenes in scene-selective regions were best explained by a DNN feature model, with no discernible unique contribution of functions. Thus, although parts of variance in the multi-arrangement behavior were captured by the DNN feature model - and this part of the behavior was reflected in the scene-selective cortex - there are clearly aspects of scene categorization behavior that were not reflected in the activity of these regions. Collectively, these results thus reveal a striking dissociation between the information that is most important for behavioral scene categorization and the information that best describes representational dissimilarity of fMRI responses in regions of cortex that are thought to support scene recognition. Below, we discuss two potential explanations for this dissociation.

First, one possibility is that functions are represented outside scene-selective cortex. Our searchlight analysis indeed revealed clusters of correlations with the function model in bilateral ventral and left lateral occipito-temporal cortex. Visual inspection of these maps suggests that these clusters potentially overlap with known face- and body-selective regions such as the Fusiform Face (FFA; *Kanwisher et al., 1997*) and Fusiform Body (FBA; *Peelen and Downing, 2007*) areas on ventral surface, as well as the Extrastriate Body Area (EBA; *Downing et al., 2001*) on the lateral surface. This lateral cluster could possibly include motion-selective (*Zeki et al., 1991*; *Tootell et al., 1995*) and tool-selective (*Martin et al., 1996*) regions as well. Our results further indicated that these searchlight clusters contained distinct representations of scenes that contained *acting* bodies, and may

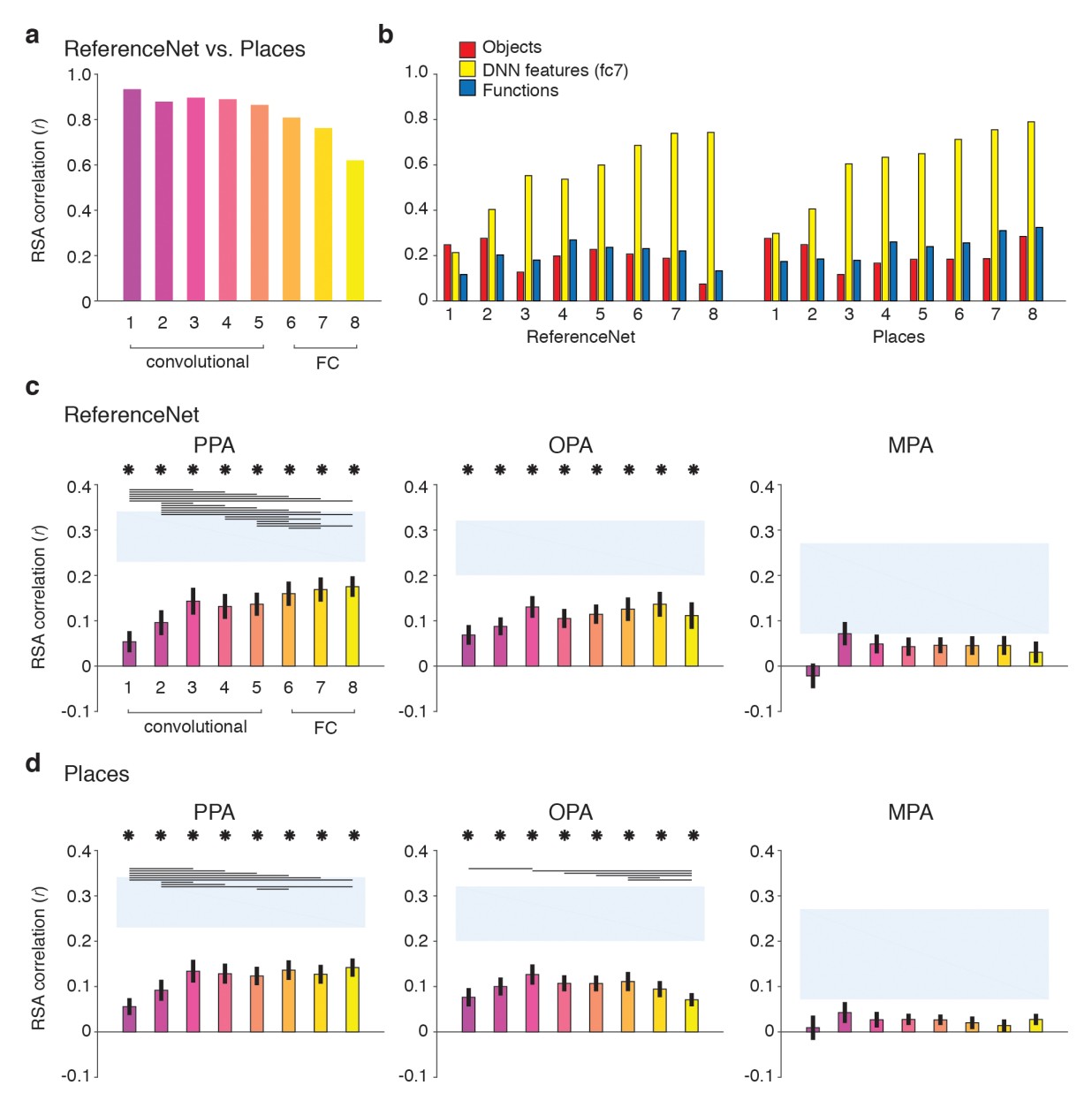

**Figure 8.** DNN layer and DNN training comparisons in terms of correlation with fMRI responses in scene-selective cortex. Panels show convolutional and fully-connected (FC) layer-by-layer RDM correlations between (A) an object-trained (ReferenceNet) and a scene-trained (Places) DNN; (B) both DNNs and the *a priori* selected feature models; (C) the object-trained DNN and scene-selective ROIs; (D) the scene-trained DNN and scene-selective ROIs (all comparisons FDR-corrected within ROI; See legend of *Figure 2B* for explanation of the statistical indicators and noise ceiling). While the decreasing correlation between DNNs indicates stronger task-specificity of higher DNN layers, the original fc7 DNN feature model correlated most strongly with high-level layers of both DNNs. The object-trained and the scene-trained DNN correlated similarly with PPA and OPA, with both showing remarkable good performance for mid-level layers. The RDMs for each individual DNN layer are provided in *Figure 1—source data 1*. Searchlight maps for each layer of the object- and scene-trained DNN are provided in *Figure 8—video 1* and *Figure 8—video 2*, respectively.

DOI: https://doi.org/10.7554/eLife.32962.010

The following videos are available for figure 8:

**Figure 8—video 1.** Layer-by-layer searchlight results for the object-trained DNN (ReferenceNet).

DOI: https://doi.org/10.7554/eLife.32962.011

**Figure 8—video 2.** Layer-by-layer searchlight results for the scene-trained DNN (Places).

DOI: https://doi.org/10.7554/eLife.32962.012

therefore partially overlap with regions important for action observation (e.g., *Hafri et al., 2017*). Lateral occipital-temporal cortex in particular is thought to support action observation by containing 'representations which capture perceptual, semantic and motor knowledge of how actions change the state of the world' (*Lingnau and Downing, 2015*). While our searchlight results suggest a possible contribution of these non-scene-selective regions to scene understanding, more research is needed to address how the functional feature model as defined here relates to the action observation network, and to what extent the correlations with functional features can be explained by mere coding of the presence of bodies and motion versus more abstract action-associated features. Importantly, the lack of a correlation between these regions and the multi-arrangement behavior suggests that these regions do not fully capture the representational space that is reflected in the function model.

The second possible explanation for the dissociation between brain and behavioral data is that the task performed during fMRI did not engage the same mental processes that participants employed during the two behavioral tasks we investigated. Specifically, both the multi-arrangement task used here and the online same-different behavioral paradigm used in (*Greene et al., 2016*) required participants to directly compare simultaneously presented scenes, while we employed a 'standard' fixation task in the scanner to prevent biasing our participants towards one of our feature models. Therefore, one possibility is that scene functions only become relevant for scene categorization when participants are engaged in a *contrastive* task, that is explicitly comparing two scene exemplars side-by-side (as in *Greene et al., 2016*) or within the context of the entire stimulus set being present on the screen (as in our multi-arrangement paradigm). Thus, the fMRI results might change with an explicit contrastive task in which multiple stimuli are presented at the same time, or with a task that explicitly requires participants to consider functional aspects of the scenes. Although we investigated one possible influence of task in the scanner by using a covert naming task in Experiment 2, resulting in deeper and more conceptual processing, it did not result in a clear increase in the correlation with the function model in scene-selective cortex. The evidence for task effects on fMRI responses in category-selective cortex is somewhat mixed: Task differences have been reported to affect multi-voxel pattern activity in both object-selective (*Harel et al., 2014*) and scene-selective cortex (*Lowe et al., 2016*), but other studies suggest that task has a minimal influence on representation in ventral stream regions, instead being reflected in fronto-parietal networks (*Erez and Duncan, 2015*; *Bracci et al., 2017*; *Bugatus et al., 2017*). Overall, our findings suggest that not all the information that contributes to scene categorization is reflected in scene-selective cortex activity 'by default', and that explicit task requirements may be necessary in order for this information to emerge in the neural activation patterns in these regions of cortex.

Importantly, the two explanations outlined above are not mutually exclusive. For example, it is possible that a task instruction to explicitly label the scenes with potential actions will activate components of both the action observation network (outside scene-selective cortex) as well as task-dependent processes within scene-selective cortex. Furthermore, given reports of potentially separate scene-selective networks for memory versus perception (*Baldassano et al., 2016*; *Silson et al., 2016*), it is likely that differences in mnemonic demands between tasks may have an important influence on scene-selective cortex activity. Indeed, memory-based navigation or place recognition tasks (*Epstein et al., 2007*; *Marchette et al., 2014*) have been shown to more strongly engage the medial parietal cortex and MPA. In contrast, our observed correlation with DNN features seems to support a primary role for PPA and OPA in bottom-up visual scene analysis, and fits well with the growing literature showing correspondences between extrastriate cortex activity and DNN features (*Cadieu et al., 2014*; *Khaligh-Razavi and Kriegeskorte, 2014*; *Güçlü and van Gerven, 2015*; *Cichy et al., 2016*; *Horikawa and Kamitani, 2017*; *Wen et al., 2017*). Our analyses further showed that DNN correlations with scene-selective cortex were not exclusive to higher DNN layers, but already emerged at earlier layers, suggesting that the neural representation in PPA/OPA may be driven more by visual features than semantic information (*Watson et al., 2017*).

One limitation of our study is that we did not exhaustively test all possible DNN models. While our design - in which we explicitly aimed to minimize inherent correlations between the feature models - required us to 'fix' the DNN features to be evaluated beforehand, many more variants of DNN models have been developed, consisting of different architectures such as VGG, GoogleNet and ResNet (*Garcia-Garcia et al., 2017*), as well as different training regimes. Here, we explored the effect of DNN training by comparing the feature representations between an object- versus a place-

trained DNN, but we did not observe strong differences in terms of their ability to explain fMRI responses in either scene-selective cortex or other parts of the brain (see whole-brain searchlights for the two DNNs in *Figure 8—video 1* and *Figure 8—video 2*). However, this does not exclude the possibility that other DNNs will map differently onto brain responses, and possibly also explain more of the behavioral measures of human scene categorization. For example, a DNN trained on the Atomic Visual Actions (AVA) dataset (*Gu et al., 2017*), or the DNNs developed in context of event understanding (e.g., the Moments in Time Dataset; *Monfort et al., 2018*) could potentially capture more of the variance explained by the function model in the scene categorization behavior. To facilitate the comparison of our results with alternative and future models, we have made the fMRI and the behavioral data reported in this paper publicly available in *Figure 1—source data 1*.

These considerations highlight an important avenue for future research in which multiple feature models (including DNNs that vary by training and architecture) and brain and behavioral measurements are carefully compared. However, our current results suggest that when participants perform scene categorization, either explicitly (*Greene et al., 2016*) or within a multi-arrangement paradigm (*Kriegeskorte and Mur, 2012*), they incorporate information that is not reflected in either the DNNs or in PPA and OPA. Our results thus highlight a significant gap between the information that is captured in both scene–selective cortex and a set of commonly used off-the-shelf DNNs, relative to the information that drives human understanding of visual environments. Visual environments are highly multidimensional, and scene understanding encompasses many behavioral goals, including not just visual object or scene recognition, but also navigation and action planning (*Malcolm et al., 2016*). While visual or DNN features likely feed into multiple of these goals - for example, by signaling navigable paths in the environment (*Bonner and Epstein, 2017*), or landmark suitability (*Troiani et al., 2014*) - it is probably not appropriate to think about the neural representations relevant to all these different behavioral goals as being contained within one single brain region or a single neural network model. Ultimately, unraveling the neural coding of scene information will require careful manipulations of both multiple tasks and multiple scene feature models, as well as a potential expansion of our focus on a broader set of regions than those characterized by the presence of scene-selectivity.

## Summary and conclusion

We successfully disentangled the type of information represented in scene-selective cortex: out of three behaviorally relevant feature models, only one provided a robust correlation with activity in scene-selective cortex. This model was derived from deep neural network features from a widely used computer vision algorithm for object and scene recognition. Intriguingly, however, the DNN model was not sufficient to explain scene categorization behavior, which was characterized by an additional strong contribution of functional information. This highlights a limitation of current DNNs in explaining scene understanding, as well as a potentially more distributed representation of scene information in the human brain beyond scene-selective cortex.

## Materials and methods

### Participants

Twenty healthy participants (13 female, mean age 25.4 years, SD = 4.6) completed the first fMRI experiment and subsequent behavioral experiment. Four of these participants (three female, mean age 24.3 years, SD = 4.6) additionally participated in the second fMRI experiment, as well as four new participants (two female, mean age 25 years, SD = 1.6), yielding a total of eight participants in this experiment. Criteria for inclusion were that participants had to complete the entire experimental protocol (i.e., the fMRI scan and the behavioral experiment). Beyond the participants reported, three additional subjects were scanned but behavioral data was either not obtained or lost. Four additional participants did not complete the scan session due to discomfort or technical difficulties. All participants had normal or corrected-to-normal vision and gave written informed consent as part of the study protocol (93 M-0170, NCT00001360) prior to participation in the study. The study was approved by the Institutional Review Board of the National Institutes of Health and was conducted according to the Declaration of Helsinki.

## MRI acquisition

Participants were scanned on a research-dedicated Siemens 7T Magnetom scanner in the Clinical Research Center on the National Institutes of Health Campus (Bethesda, MD). Partial T2*-weighted functional image volumes were acquired using a gradient echo planar imaging (EPI) sequence with a 32-channel head coil (47 slices; 1.6 × 1.6×1.6 mm; 10% interslice gap; TR, 2 s; TE, 27 ms; matrix size, 126 × 126; FOV, 192 mm). Oblique slices were oriented approximately parallel to the base of the temporal lobe and were positioned such that they covered the occipital, temporal, and parietal cortices, and as much as possible of frontal cortex. After the functional imaging runs, standard MPRAGE (magnetization-prepared rapid-acquisition gradient echo) and corresponding GE-PD (gradient echo–proton density) images were acquired, and the MPRAGE images were then normalized by the GE-PD images for use as a high-resolution anatomical image for the following fMRI data analysis (*Van de Moortele et al., 2009*).

## Stimuli and models

Experimental stimuli consisted of color photographs of real-world scenes (256 × 256 pixels) from 30 scene categories that were selected from a larger image database previously described in (*Greene et al., 2016*). These scene categories were picked using an iterative sampling procedure that minimized the correlation between the categories across three different models of scene information: functions, object labels and DNN features, with the additional constraint that the final stimulus set should have equal portions of categories from indoor, outdoor man-made and outdoor natural scenes, which is the largest superordinate distinction present in the largest scene-database that is publicly available, the SUN database (*Xiao et al., 2014*). As obtaining a guaranteed minimum was impractical, we adopted a variant of the odds algorithm (*Bruss, 2000*) as our stopping rule. Specifically, we created 10,000 sets of 30 categories and measured the correlations between functional, object, and DNN RDMs (distance metric: Spearman's *rho*), noting the minimal value from the set. We persisted in this procedure until we observed a set with lower inter-feature correlations than was observed in the initial 10,000. From each of the final selected scene categories, eight exemplars were randomly selected and divided across two separate stimulus sets of 4 exemplars per scene category. Stimulus sets were assigned randomly to individual participants (Experiment 1: stimulus set 1, n = 10; stimulus set 2, n = 10; Experiment 2, stimulus set 1, n = 5; stimulus set 2, n = 3). Participants from Experiment 2 that had also participated in Experiment 1 were presented with the other stimulus set than the one they saw in Experiment 1.

## fMRI procedure

Participants were scanned while viewing the stimuli on a back-projected screen through a rear-view mirror that was mounted on the head coil. Stimuli were presented at a resolution of 800 × 600 pixels such that they subtended ~10×10 degrees of visual angle. Individual scenes were presented in an event-related design for a duration of 500 ms, separated by a 6 s interval. Throughout the experimental run, a small fixation cross (<0.5 degrees) was presented in the center of the screen.

In Experiment 1, participants performed a task on the central fixation cross that was unrelated to the scenes. Specifically, simultaneous with the presentation of each scene, either the vertical or horizontal arm of the fixation cross became slightly elongated and participants indicated which arm was longer by pressing one of two buttons indicated on a hand-held button box. Both arms changed equally often within a given run and arm changes were randomly assigned to individual scenes. In Experiment 2, the fixation cross had a constant size, and participants were instructed to covertly name the scene whilst simultaneously pressing one button on the button box. To assure that participants in Experiment 2 were able to generate a name for each scene, they were first familiarized with the stimuli. Specifically, prior to scanning, participants were presented with all scenes in the set in randomized order on a laptop in the console room. Using a self-paced procedure, each scene was presented in isolation on the screen accompanied by the question 'How would you name this scene?'. The participants were asked to type one or two words to describe the scene; as they typed, their answer appeared under the question, and they were able to correct mistakes using backspace. After typing the self-generated name, participants hit enter and the next scene would appear until all 120 scenes had been seen by the participant. This procedure took about ~10 min.

In both Experiment 1 and 2, participants completed eight experimental runs of 6.4 min each (192 TRs per run); one participant from Experiment 1 only completed seven runs due to time constraints. Each run started and ended with a 12 s fixation period. Each run contained two exemplar presentations per scene category. Individual exemplars were balanced across runs such that all stimuli were presented after two consecutive runs, yielding four presentations per exemplar in total. Exemplars were randomized across participants such that each participant always saw the same two exemplars within an individual run; however the particular combination was determined anew for each individual participant and scene category. Stimulus order was randomized independently for each run. Stimuli were presented using PsychoPy v1.83.01 (*Peirce, 2007*).

## Functional localizers

Participants additionally completed four independent functional block-design runs (6.9 min, 208 TRs) that were used to localize scene-selective regions of interest (ROIs). Per block, twenty gray-scale images (300 × 300 pixels) were presented from one of eight different categories: faces, man-made and natural objects, buildings, and four different scene types (man-made open, man-made closed, natural open, natural closed; *Kravitz et al., 2011*) while participants performed a one-back repetition-detection task. Stimuli were presented on a gray background for 500 ms duration, separated by 300 ms gaps, for blocks of 16 s duration, separated by 8 s fixation periods. Categories were counterbalanced both within runs (such that each category occurred twice within a run in a mirror-balanced sequence) and across runs (such that each category was equidistantly spaced in time relative to each other category across all four runs). Two localizer runs were presented after the first four experimental runs and two after the eight experimental runs were completed but prior to the T1 acquisition. For four participants, only two localizer runs were collected due to time constraints.

## Behavioral experiment

On a separate day following the MRI data acquisition, participants performed a behavioral multi-arrangement experiment. In a behavioral testing room, participants were seated in front of a desktop computer with a Dell U3014 monitor (30 inches, 2560 x 1600 pixels) on which all 120 stimuli that the participant had previously seen in the scanner were displayed as thumbnails around a white circular arena. A mouse-click on an individual thumbnail displayed a larger version of that stimulus in the upper right corner. Participants were instructed to arrange the thumbnails within the white circle in such a way that the arrangement would reflect 'how similar the scenes are, whatever that means to you', by means of dragging and dropping the individual exemplar thumbnails. We purposely avoided providing specific instructions in order to not bias participants towards using either functions, objects or DNN features to determine scene similarity. Participants were instructed to perform the task at their own pace; if the task took longer than 1 hr, they were encouraged to finish the experiment (almost all participants took less time, averaging a total experiment duration of ~45 mins). Stimuli were presented using the MATLAB code provided in (*Kriegeskorte and Mur, 2012*). To obtain insight in the sorting strategies used by participants, they were asked (after completing the experiment) to take a few minutes to describe how they organized the scenes, using a blank sheet of paper and a pen, using words, bullet-points or drawings.

## Behavioral data analysis

Behavioral representational dissimilarity matrices (RDMs) were constructed for each individual participant by computing the pairwise squared on-screen distances between the arranged thumbnails and averaging the obtained distances across the exemplars within each category. The relatedness of the models and the behavioral data was determined in the same manner as for the fMRI analysis, that is by computing both individual model correlations and unique and shared variance across models via hierarchical regression (see below).

## fMRI preprocessing

Data were analyzed using AFNI software (https://afni.nimh.nih.gov). Before statistical analysis, the functional scans were slice-time corrected and all the images for each participant were motion corrected to the first image of the first functional run after removal of the first and last six TRs from

each run. After motion correction, the localizer runs were smoothed with a 5 mm full-width at half-maximum Gaussian kernel; the event-related data was not smoothed.

## fMRI statistical analysis: localizers

Bilateral ROIs were created for each individual participant based on the localizer runs by conducting a standard general linear model implemented in AFNI. A response model was built by convolving a standard gamma function with a 16 s square wave for each condition and compared against the activation time courses using Generalized Least Squares (GLSQ) regression. Motion parameters and four polynomials accounting for slow drifts were included as regressors of no interest. To derive the response magnitude per category, t-tests were performed between the category-specific beta estimates and baseline. Scene-selective ROIs were generated by thresholding the statistical parametric maps resulting from contrasting scenes > faces at p<0.0001 (uncorrected). Only contiguous clusters of voxels (>25) exceeding this threshold were then inspected to define scene-selective ROIs consistent with previously published work (*Epstein, 2005*). For participants in which clusters could not be disambiguated, the threshold was raised until individual clusters were clearly identifiable. While PPA and OPA were identified in all participants for both Experiment 1 and 2, MPA/RSC was detected in only 14 out 20 participants in Experiment 1, and all analyses for this ROI in Experiment 1 are thus based on this subset of participants.

## fMRI statistical analysis: event-related data

Each event-related run was deconvolved independently using the standard GLSQ regression model in AFNI. The regression model included a separate regressor for each of the 30 scene categories as well as motion parameters and four polynomials to account for slow drifts in the signal. The resulting beta-estimates were then used to compute representational dissimilarity matrices (RDMs; (*Kriegeskorte et al., 2008*) based on the multi-voxel patterns extracted from individual ROIs. Specifically, we computed pairwise cross-validated Mahalanobis distances between each of the scene 30 categories following the approach outlined in (*Walther et al., 2016*). First, multi-variate noise normalization was applied by normalizing the beta-estimates by the covariance matrix of the residual time-courses between voxels within the ROI. Covariance matrices were regularized using shrinkage toward the diagonal matrix (*Ledoit and Wolf, 2004*). Unlike univariate noise normalization, which normalizes each voxel's response by its own error term, multivariate noise normalization also takes into account the noise covariance between voxels, resulting in more reliable RDMs (*Walther et al., 2016*). After noise normalization, squared Euclidean distances were computed between individual runs using a leave-one-run-out procedure, resulting in cross-validated Mahalanobis distance estimates. Note that unlike correlation distance measures, cross-validated distances provide unbiased estimates of pattern dissimilarity on a ratio scale (*Walther et al., 2016*), thus providing a distance measure suitable for direct model comparisons.

## Model comparisons: individual models

To test the relatedness of the three models of scene dissimilarity with the measured fMRI dissimilarity, the off-diagonal elements of each model RDM were correlated (Pearson's *r*) with the off-diagonal elements of the RDMs of each individual participant's fMRI ROIs. Following (*Nili et al., 2014*), the significance of these correlations was determined using one-sided signed-rank tests against zero, while pairwise differences between models in terms of their correlation with fMRI dissimilarity were determined using two-sided signed-ranked tests. For each test, we report the sum of signed ranks for the number of observations W(*n*) and the corresponding p-value; for tests with n > 10 we also report the z-ratio approximation. The results were corrected for multiple comparisons (across both individual model correlations and pairwise comparisons) using FDR correction (*Benjamini and Hochberg, 1995*) for each individual ROI separately. Noise ceilings were computed following (*Nili et al., 2014*): an upper bound was estimated by computing the correlation between each participant's individual RDM and the group-average RDM, while a lower bound was estimated by correlating each participant's RDM with the average RDM of the other participants (leave-one-out approach). The participant-averaged RDM was converted to rank order for visualization purposes only.

## Model comparisons: partial correlations and variance partitioning

To determine the contribution of each individual model when considered in conjunction with the other models, we performed two additional analyses: partial correlations, in which each model was correlated (Pearsons $r$) while partialling out the other two models, and variance partitioning based on multiple linear regression. For the latter, the off-diagonal elements of each ROI RDM were assigned as the dependent variable, while the off-diagonal elements of the three model RDMs were entered as independent variables (predictors). To obtain unique and shared variance across the three models, seven multiple regression analyses were run in total: one 'full' regression that included all three feature models as predictors; and six reduced models that included as predictors either combinations of two models in pairs (e.g., functions and objects), or including each model by itself. By comparing the explained variance ($r^2$) of a model used alone to the $r^2$ of that model in conjunction with another model, we can infer the amount of variance that is independently explained by that model, that is partition the variance (see *Groen et al., 2012*; *Ramakrishnan et al., 2014*; *Lescroart et al., 2015*; *Çukur et al., 2016*; *Greene et al., 2016*; *Hebart et al., 2018* for similar approaches).

Analogous to the individual model correlation analyses, partial correlations were calculated for each individual participant separately, and significance was determined using one-sided signed-rank tests across participants (FDR-corrected across all comparisons within a given ROI). To allow comparison with the results reported in (*Greene et al., 2016*), variance partitioning was performed on the participant-average RDMs. Similar results were found, however, when variance was partitioned for individual participant's RDMs and then averaged across participants. To visualize this information in an Euler diagram, we used the EulerAPE software (*Micallef and Rodgers, 2014*).

## Variance partitioning of fMRI based on models and behavior

Using the same approach as in the previous section, a second set of regression analyses was performed to determine the degree of shared variance between the behavioral categorization on the one hand, and the functions and DNN features on the other, in terms of the fMRI response pattern dissimilarity. The Euler diagrams were derived using the group-average RDMs, taking the average result of the multi-arrangement task of these participants as the behavioral input into the analysis.

## Direct reproducibility test of representational structure in behavior and fMRI

To assess how well the obtained RDMs were reproducible within each measurement domain (behavior and fMRI), we compared the average RDMs obtained for the two separate stimulus sets. Since these two sets of stimuli were viewed by different participants (see above under 'Stimuli and models'), this comparison provides a strong test of generalizability, across both scene exemplars and across participant pools. Set-average RDMs were compared by computing inter-RDM correlations (Pearson's $r$) and 96% confidence intervals (CI) and statistically tested for reproducibility using a random permutation test based on 10.000 randomizations of the category labels.

## DNN comparisons

The original, *a priori* fc7 DNN feature model was determined based on the large set of exemplars (average of 65 exemplars per scene category) used in *Greene et al. (2016)*. To investigate the influence of DNN layer and training images on the learned visual features and their correspondence with activity in scene-selective cortex, we derived two new sets of RDMs by passing our scene stimuli through two pre-trained, 8-layer AlexNet (*Krizhevsky et al., 2017*) architecture networks: (1) a 1000-object label ImageNet-trained (*Deng et al., 2009*) network implemented in Caffe (*Jia et al., 2014*) ('ReferenceNet') and (2) a 250-scene label Places-trained network ('Places') (*Zhou et al., 2014*). By extracting the node activations from each layer, we computed pairwise dissimilarity (1 – Pearson's $r$) resulting in one RDM per layer and per model. These RDMs were then each correlated with the fMRI RDMs from each participant in PPA, OPA and MPA (Pearson's $r$). These analyses were performed on the combined data of Experiment 1 and 2; RDMs for participants that participated in both Experiments (n = 4) were averaged prior to group-level analyses.

## Searchlight analyses

To test the relatedness of functions, objects and visual feature models with fMRI activity recorded outside scene-selective ROIs, we conducted whole-brain searchlight analyses. RDMs were computed in the same manner as for the ROI analysis, that is by computing cross-validated Mahalanobis distances based on multivariate noise-normalized multi-voxel patterns, but now within spherical ROIs of 3 voxel diameter (i.e. 123 voxels/searchlight). Analogous to the ROI analyses, we computed partial correlations of each feature model, correcting for the contributions of the remaining two models. These partial correlation coefficients were assigned to the center voxel of each searchlight, resulting in one whole-volume map per model. Partial correlation maps were computed for each participant separately in their native volume space. To allow comparison at the group level, individual participant maps were first aligned to their own high-resolution anatomical scan and then to surface reconstructions of the grey and white matter boundaries created from these high-resolution scans using the Freesurfer (http://surfer.nmr.mgh.harvard.edu/) 5.3 autorecon script using SUMA (Surface Mapping with AFNI) software (https://afni.nimh.nih.gov/Suma). The surface images for each participant were then smoothed with a Gaussian 10 mm FWHM filter in surface coordinate units using the SurfSmooth function with the HEAT_07 smoothing method.

Group-level significance was determined by submitting these surface maps to node-wise one-sample t-tests in conjunction with Threshold Free Cluster Enhancement (*Smith and Nichols, 2009*) through Monte Carlo simulations using the algorithm implemented in the CoSMoMVPA toolbox (*Oosterhof et al., 2016*), which performs group-level comparisons using sign-based permutation testing (n = 10,000) to correct for multiple comparisons. To increase power, the data of Experiment 1 and 2 were combined; coefficient maps for participants that participated in both Experiments (n = 4) were averaged prior to proceeding to group-level analyses.

For searchlight comparisons with scene categorization behavior and feature models based on different DNN layers, we computed regular correlations (Pearson's *r*) rather than partial correlations. For the behavioral searchlight, we used the average multi-arrangement behavior from Experiment 1 (since the participants from Experiment 2 did not perform this task). For the DNN searchlights, we used the same layer-by-layer RDMs as for the ROI analyses, independently correlating those with the RDMs of each spherical ROI. Group-level significance was determined in the same manner as for the *a priori* selected feature models (see above).

## Acknowledgements

This work was supported by the Intramural Research Program (ZIAMH002909) of the National Institutes of Health – National Institute of Mental Health Clinical Study Protocol 93 M-0170, NCT00001360. IIAG was also supported by a Rubicon Fellowship from the Netherlands Organization for Scientific Research (NWO). LF and DMB were funded by the Office of Naval Research Multidisciplinary University Research Initiative Grant N000141410671.

## Additional information

### Funding

| Funder | Grant reference number | Author |
|---|---|---|
| National Institutes of Health | ZIAMH002909 | Iris IA Groen<br>Chris I Baker |
| Nederlandse Organisatie voor Wetenschappelijk Onderzoek | Rubicon Fellowship | Iris IA Groen |
| Office of Naval Research | Multidisciplinary Research Initiative Grant N000141410671 | Li Fei-Fei<br>Diane M Beck |

The funders had no role in study design, data collection and interpretation, or the decision to submit the work for publication.

## Author contributions
Iris IA Groen, Conceptualization, Data curation, Software, Formal analysis, Funding acquisition, Validation, Investigation, Visualization, Methodology, Writing—original draft; Michelle R Greene, Conceptualization, Resources, Software, Methodology, Writing—review and editing; Christopher Baldassano, Conceptualization, Resources, Methodology, Writing—review and editing; Li Fei-Fei, Diane M Beck, Conceptualization, Supervision, Funding acquisition, Writing—review and editing; Chris I Baker, Conceptualization, Resources, Supervision, Funding acquisition, Project administration, Writing—review and editing

## Author ORCIDs
Iris IA Groen http://orcid.org/0000-0002-5536-6128
Christopher Baldassano http://orcid.org/0000-0003-3540-5019
Diane M Beck http://orcid.org/0000-0001-9802-5828
Chris I Baker http://orcid.org/0000-0001-6861-8964

## Ethics
Human subjects: All participants had normal or corrected-to-normal vision and gave written informed consent as part of the study protocol (93 M-0170, NCT00001360) prior to participation in the study. The study was approved by the Institutional Review Board of the National Institutes of Health and was conducted according to the Declaration of Helsinki.

## Decision letter and Author response
Decision letter https://doi.org/10.7554/eLife.32962.015
Author response https://doi.org/10.7554/eLife.32962.016

# Additional files

## Supplementary files
• Transparent reporting form
DOI: https://doi.org/10.7554/eLife.32962.013

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
