## [Decision Letter]

Thank you for submitting your article "Distinct contributions of functional and deep neural network features to scene representation in brain and behavior" for consideration by *eLife*. Your article has been reviewed by two peer reviewers, and the evaluation has been overseen by a Reviewing Editor and Timothy Behrens as the Senior Editor. The reviewers have opted to remain anonymous.

The reviewers have discussed the reviews with one another and the Reviewing Editor has drafted this decision to help you prepare a revised submission.

Summary:

The manuscript compares how three models of scene properties explain human behavior (scene similarity judgments) and fMRI activation patterns in scene-selective regions. Behavior can be explained by function/action labels and by features of a deep neural network (DNN). Brain activation patterns, on the other hand, were mainly explained by the DNN features, with little contribution from function/action features (or from the third model, based on object features). The authors emphasize the apparent distinction between behavior and brain activation patterns in their use of functional features.

Overall, this is a very clearly written, straightforward paper. The paper is informative and useful in understanding both the general organization of visual processing brain regions, and the relation between DNNs and human brains. The approach of pre-selecting images to minimize intrinsic correlations between the three feature spaces is clever. There's novelty in how they describe unique contributions of each models: in addition to a conventional RSA correlation, the paper reports results of a variance partitioning analysis along with Euler diagram visualization which allows readers to easily see independent and shared contribution of each models. Experiment 2 logically follows questions from Experiment 1, such as whether the discrepancy between behavioral and fMRI results comes from task differences.

Essential revisions:

1) My main concern was that the authors tend to over-generalize and overstate their results. The results will remain interesting when presented at face value, and the description will be much more accurate.

In an apparent effort to hype up the results (or maybe to keep the conclusions compatible with the previous related paper by Greene, 2016?), the authors insist that functional features explain behavior while DNN features explain brain activity. But in fact, DNN features explain a very large part of behavior as well. It would seem more accurate to conclude that both functional and DNN features explain behavior. As functional features do not contribute to brain activation patterns in scene regions (but DNN features do), there is still an interesting dissociation.

2) The authors talk about "behavior" as if the results could be generalized to any scene categorization behavior. But in practice, the functional (or DNN) features only explain a very specific type of behavior, a scene multi-arrangement similarity sorting task. This limitation is addressed a bit in the Discussion, but I would feel much more comfortable if the authors could systematically qualify their statements about behavior (in the title, Abstract, Results).

3) Similarly, DNN features are referred to as if that was a unique, unambiguous "thing", but there are many types of DNN features depending on network architecture, training data, and hierarchical layer. The authors address the latter two factors to some extent, by reporting correlations with all 8 layers of AlexNet, and with another version of AlexNet trained on places rather than objects. But these are not the only two possibilities. In particular, it seems plausible that a deep network trained on functional (or action) labels for each scene (like the AVA dataset or the YouTube-8M dataset) could behave very differently, and possibly explain more of the behavioral data. In addition, there are many more (and more advanced) image categorization architectures than AlexNet (VGG, Inception, ResNet, etc.). Talking about DNNs "in general" based on data from AlexNet alone is like talking about "mammals" based on mouse data alone. I am not suggesting that the study should include other DNNs, but rather that the descriptions and conclusions should more accurately reflect the actual experimental manipulations.

4) A mild concern is that some of the analysis and interpretations seem too post-hoc. In other words, a theoretical motivation for choosing the three models is not explained enough, other than that these are the models used in a previous study from the same group. These models are chosen with an assumption that feature spaces that were relevant for a behavioral scene study should be relevant for brain responses, which seems natural but creates a disadvantage that each models are not explained/introduced enough to understand why these models may be relevant for behavior and neural representation of scenes. These models deserve more background in the introduction. Are there any alternative models that might contribute which is not tested in their previous study? Does the scene categorization task (similarity sorting) used in behavioral task relates to what the visual system might naturally do while perceiving real world scenes?

5) Overall conclusion and a theoretical take home point seem relatively weak. It is clear that there is a discrepancy between behavioral and neural representations of scenes. What does this tell us about neural representations in scene cortex? For example, a discussion about what exactly different layers of DNN that contributes to neural response is/might be representing should help. It is vaguely mentioned that mid and high level DNN layers play important roles, with specific names of DNN layer, e.g., fc7. However, this doesn't clarify what features are represented in these layers and how it corresponds to different processes involved in scene perception/recognition. Richer introduction and discussion about DNN should help readers understand the theoretical contribution of this work on what and how features are represented in scene perception.

6) What does the current data tell us about existing behavioral findings by Greene et al., 2017? Should the contribution of functional model be interpreted differently and specifically for the behavioral scene categorization task or the types of images (e.g., images with humans/actions) used in the previous research? If not, that should be discussed too. In other words, there needs to be bi-directional discussion between what current discrepancy results inform us about behavioral and neural findings about scene representation.

Minor points [abridged]:

[…] 7.Doesn't the absence of DNN feature space similarity with brain activity (outside of the two small ventral and lateral occipitotemporal clusters) in the searchlight analysis contradict previous findings by Khaligh-Razavi & Kriegeskorte or Guclu & van Gerven? Shouldn't that be discussed explicitly?

---

## [Author Response]

Essential revisions:1) My main concern was that the authors tend to over-generalize and overstate their results. The results will remain interesting when presented at face value, and the description will be much more accurate.In an apparent effort to hype up the results (or maybe to keep the conclusions compatible with the previous related paper by Greene, 2016?), the authors insist that functional features explain behavior while DNN features explain brain activity. But in fact, DNN features explain a very large part of behavior as well. It would seem more accurate to conclude that both functional and DNN features explain behavior. As functional features do not contribute to brain activation patterns in scene regions (but DNN features do), there is still an interesting dissociation.

Our findings indeed indicate that DNN features also contribute unique variance to behavior, and a similar result was reported in Greene et al., (2016). In the current manuscript, we highlight that the unique variance explained by the functional features is (numerically) larger than that explained by the DNN features (see Figure 2), and that the functional feature space is also more often the best-fitting model in individual subjects (see Figure 2), because we wanted to emphasize that this ‘ranking’ of the feature spaces constitutes a direct replication of the earlier study which used a much larger stimulus set and a different experimental task and subject population. Nevertheless, we did not intend to sweep the contribution of the DNNs under the rug. In fact, we think this finding constitutes an important validation of the increased popularity of the application of DNNs in neuroscience experiments. Therefore, we have amended our statements throughout the manuscript to better reflect the conclusion that behavior is explained by both functional features and DNNs (see Abstract, Introduction, Results and Materials and methods).

2) The authors talk about "behavior" as if the results could be generalized to any scene categorization behavior. But in practice, the functional (or DNN) features only explain a very specific type of behavior, a scene multi-arrangement similarity sorting task. This limitation is addressed a bit in the Discussion, but I would feel much more comfortable if the authors could systematically qualify their statements about behavior (in the title, Abstract, Results).

We agree with the reviewer that using ‘behavior’ as a blanket term is too broad. We have therefore amended our title, Abstract and Results and Discussion sections to more precisely indicate the type of behavior we are referring to, now using the term ‘multi-arrangement’, ‘similarity assessment’ or ‘similarity judgments’ for the behavioral data we present here (see title page and Abstract, Introduction, Results and Materials and methods). However, we do think it is important to emphasize that the behavioral similarity assessments obtained in this paper with the multi-arrangement task are consistent with the results obtained with an explicit same-different categorization task in Greene et al., (2016), and that those assessments also correlate with responses in scene-selective regions (see Figure 4/B). This suggests that the multi-arrangement task does capture elements of perhaps more general scene categorization behavior.

3) Similarly, DNN features are referred to as if that was a unique, unambiguous "thing", but there are many types of DNN features depending on network architecture, training data, and hierarchical layer. The authors address the latter two factors to some extent, by reporting correlations with all 8 layers of AlexNet, and with another version of AlexNet trained on places rather than objects. But these are not the only two possibilities. In particular, it seems plausible that a deep network trained on functional (or action) labels for each scene (like the AVA dataset or the YouTube-8M dataset) could behave very differently, and possibly explain more of the behavioral data. In addition, there are many more (and more advanced) image categorization architectures than AlexNet (VGG, Inception, ResNet, etc.). Talking about DNNs "in general" based on data from AlexNet alone is like talking about "mammals" based on mouse data alone. I am not suggesting that the study should include other DNNs, but rather that the descriptions and conclusions should more accurately reflect the actual experimental manipulations.

We agree that a DNN trained on a different type of scene label may result in a different outcome. Our motivation to use the ImageNet-trained AlexNet as one of our three feature models was based on it being trained on one the largest database available at the time, meaning that its features could be expected to reflect a representative sampling of the visual world. Moreover, this version was trained on an object categorization task, which previous work has suggested correlates with brain activity (e.g. Khaligh-Razavi and Kriegeskorte, 2014). Thus, our objective was not to find the ‘best’ DNN to match our behavior, but to test how a feature space previously studied in the literature compared to other feature models that were previously also shown to predict behavioral scene categorization. However, since then indeed many other networks have been published with a wide range of architectures and training data, which is why we decided to include at least one alternate DNN (Places: trained on scene categorization) in the paper, albeit as a post-hoc analysis. In addition, we now explicitly mention the possibility that future DNNs will be able to capture more of the behavioral and fMRI variance, and highlight that this is an important avenue for future research: see new Discussion paragraph (fifth). Furthermore, we have removed the statement from the Abstract that DNNs represent only a subset of behaviorally relevant scene information, since we cannot claim that we have exhaustively tested all DNNs. Importantly, this is one of the reasons we make the behavioral and fMRI data publicly available along with the manuscript, in order to facilitate future comparisons of these data against other candidate models.

4) A mild concern is that some of the analysis and interpretations seem too post-hoc. In other words, a theoretical motivation for choosing the three models is not explained enough, other than that these are the models used in a previous study from the same group. These models are chosen with an assumption that feature spaces that were relevant for a behavioral scene study should be relevant for brain responses, which seems natural but creates a disadvantage that each models are not explained/introduced enough to understand why these models may be relevant for behavior and neural representation of scenes. These models deserve more background in the introduction. Are there any alternative models that might contribute which is not tested in their previous study? Does the scene categorization task (similarity sorting) used in behavioral task relates to what the visual system might naturally do while perceiving real world scenes?

Two important points are raised here. First, the reviewer asks for clarification on the selection of models. It is important to stress that the three feature spaces used here were not the only models tested in the previous behavioral study, but that these were the best performing models out of a large range of low-to-high level scene feature spaces, including other assessments of function-like scene features (i.e., an attribute model). In Greene et al., (2016) all of these models were compared with the largest-scale assessment of scene similarity to date, namely online behavioral categorization of the entire SUN database, and unique variance was assessed for several of the top-performing models. While it remains possible that there are alternative models that contribute to scene representation (including unknown future models), we reasoned that this extensive assessment of behavioral relevance of features was an appropriate ground for pre-selecting models for a neuroimaging experiment (in which we are necessarily limited in the number of scene exemplars we can show) in an informed way. An important (if not the most important) goal of neuroscience is to uncover how neural representations subserve behavior, but this aspect has not necessarily always been as prominent in previous studies on the scene representation, where stimulus selection was often based on intuitions about what the relevant scene dimensions are, not explicit behavioral assessments. For studies that did include scene categorization behavior, the behavioral ‘reality’ of the examined feature spaces has often been assessed after rather than prior to stimulus selection (e.g. Walther et al., 2009; Kravitz et al., 2011). Following the reviewer’s request, we have elaborated the description of the previous study in the Introduction (third paragraph), now providing more context for the types of models that were tested before, and we also added the explicit statement that our goal was to test these top behavioral models.

The second point concerns the degree to which the similarity assessments in our behavioral paradigm can be said to be representative of ‘real’ scene perception behavior. This is an intriguing question. A literal interpretation of this task would imply that we are constantly engaged in categorizing or making similarity judgments of the environments we are in. This seems unlikely: performing a repeated categorization task on a second-to-second basis (“This is an office. This is an office”) does not seem like plausible behavior in real-world settings. In a similar vein, judgments of pictures on a screen can be said to be impoverished way of assessing any type of ‘real’ visual behavior, and scene understanding actually constitutes a whole set of behaviors that supersede rapid categorization of images (Malcolm et al., 2016). On the other hand, a classic view in cognitive science does argue that ‘similarity’ is a fundamental way of organizing the world (Tversky, 1977; Edelman, 1998), i.e. ‘to recognize is to categorize’. Therefore, by using a scene similarity task, we are aligning ourselves with this long-standing framework which suggests that perception occurs within *some* similarity reference frame. In our study, we ask: what are the dimensions of the natural ‘similarity space’ that the visual system operates in? Which reference frame best matches the data? We chose the free arrangement task precisely because it allows for testing multiple reference frames: participants were allowed to judge scene similarity along any dimension they see fit, rather than being forced to use image-based properties only, as might for example be the case in a rapid one-back matching task. Using a free arrangement task to assess scene similarity thus opens up the possibility of getting closer to what the visual system ‘naturally does’ when perceiving real-world scenes.

5) Overall conclusion and a theoretical take home point seem relatively weak. It is clear that there is a discrepancy between behavioral and neural representations of scenes. What does this tell us about neural representations in scene cortex? For example, a discussion about what exactly different layers of DNN that contributes to neural response is/might be representing should help. It is vaguely mentioned that mid and high level DNN layers play important roles, with specific names of DNN layer, e.g., fc7. However, this doesn't clarify what features are represented in these layers and how it corresponds to different processes involved in scene perception/recognition. Richer introduction and discussion about DNN should help readers understand the theoretical contribution of this work on what and how features are represented in scene perception.

The question what exactly is represented in different layers of DNN models is an important one. In fact, this is an active area of research in both computer vision and computational neuroscience, where various methods are currently being developed to visualize and quantify the learned feature selectivity (e.g., Zhou et al., 2014; Güçlü and van Gerven, 2015; Bau et al., 2017; more relevant papers are cited on http://netdissect.csail.mit.edu). So far, these methods suggest that while earlier layers contain local filters that resemble V1-like receptive fields, higher layers develop selectivity for entire objects or object parts, perhaps resembling category-selective regions in visual cortex. We now clarify the role of different layers in the Results section that accompanies the layer-by-layer analyses in Figure 8 (see subsection “Scene-selective cortex correlates with features in both mid- and high-level DNN layers”). However – not unlike our understanding of intermediate visual areas in the brain – the exact representations of features in mid-level DNN layers remains somewhat elusive (but see Bonner and Epstein (2017) for an interesting exploration of this in the context of scene representation). This ‘black-box’ characteristic is one of the reasons why the utility of DNNs for understanding neural representation is debated (Kay, 2017; Scholte, 2018; Tripp, 2018; Turner et al., 2018). We now include more discussion of DNNs in the new Discussion paragraph (fifth paragraph) where we also discuss the possibility of testing other DNNs, in response to suggestions under point 3 above.

While we think the nature of DNN representations is a relevant and interesting question, the main focus of our paper is on another problem, which applies to any model of scene representation including DNNs, namely that model features are likely correlated with many other model features (or layers), which makes it difficult to assign a unique role to a specific model (or layer). The general assumption in the field is that because DNNs achieve classification performance at human-like levels, the representations in the DNN layers must be relevant for human behavior. While reports of superior correlations between DNN representations and neural measurement relative to other models of image representation (Yamins et al., 2013; Cadieu et al., 2014; Khaligh-Razavi and Kriegeskorte, 2014) indeed seem to support the relevance of DNNs for ‘biological’ perception, we here tested this hypothesis carefully in the context of other known behaviorally relevant feature spaces, asking to what extent the DNN features *uniquely* contribute not only to brain responses but also to scene categorization behavior. Our results show that a significant correlation between DNN features and the brain does not imply that DNNs are able to capture all elements of scene categorization behavior. Thus, while the theoretical implications of our results are indeed not easily summarized in a single take home point, we think our study highlights important limitations of focusing only on the brain-DNN relationship without simultaneously taking into account human categorization behavior. At the same time, we provide a methodological approach (variance partitioning) that might help better tease which DNN features contribute unique variance between layers, for example.

6) What does the current data tell us about existing behavioral findings by Greene et al., 2017? Should the contribution of functional model be interpreted differently and specifically for the behavioral scene categorization task or the types of images (e.g., images with humans/actions) used in the previous research? If not, that should be discussed too. In other words, there needs to be bi-directional discussion between what current discrepancy results inform us about behavioral and neural findings about scene representation.

We do not think the current findings should lead to a different interpretation of the findings in Greene et al., (2016). As highlighted in our response to points 1 and 2 above, we replicated the main finding from that study with a much-reduced stimulus set, an explicit minimization of correlation with other feature spaces, a different participant population (online workers vs. lab participants), and a different task paradigm (side-by-side categorization vs. free arrangement). Based on our exploratory result of a searchlight correlation of the function feature space with parts of cortex that exhibit potential overlap with body- or motion-selective regions, we raised the possibility that some of the functional feature space dimensions correlate with the presence of acting humans or implied motion in a subset of the scene categories. However, this is less likely to be an explanation of the results in Greene et al., (2016) because the set of images used there was much larger and was also explicitly compared to many other feature spaces. Moreover, the lack of a correlation with behavior in these regions (see below) suggests that the variance captured by the function model there may not be the same variance that drives the relationship of the function model with the behavior. As highlighted in the second and third paragraphs of the Discussion, one explanation for the discrepancy between the behavioral and fMRI findings is the experimental circumstances under which the functional features become ‘activated’. The lack of correlation between fMRI responses and the functional model suggests that brain representations of functions may be partly task-driven, happening only when participants are engaged in a contrastive task.

We agree that the question in the other direction, i.e. whether previous neural findings on scene-selective regions should be interpreted differently based on our results, is also relevant. Previous studies have reported correlations between fMRI responses in for example PPA and scene categorization behavior (e.g. Walther et al., 2009; Kravitz et al., 2011). Our results are not inconsistent with these reports: we also find that PPA responses correlate with behavior (see Figure 4 and new Figure 7). In addition, we show that DNN features likely underlie this correlation, since these features correlate both with behavior and PPA (see Figure 2/E, Figure 3/C, and Figure 8). Had we considered *only* this feature model, then we would have been able to present a ‘full circle’ account that confirms the importance of scene-selective cortex for scene perception. However, both the Greene et al., (2016) study and our current findings indicate that DNN features do not fully account for scene perception, because functional features are also (and perhaps even more) important for scene categorization behavior. Of course, it could have been the case that these features were also represented in PPA, or other scene-selective regions. However, our fMRI results clearly show that they were not, at least not under the experimental conditions we considered. Thus, apart from suggesting that neural representations relevant for scene categorization behavior may be partly task-driven (see above), these findings also raise questions about whether scene categorization should be thought of as relating exclusively to scene-selective cortex. We bring up this possibility in multiple places in the Discussion, but since our results are mainly exploratory in this regard, we think it is better not to speculate much further based on this particular study.

Minor points [abridged]:[…] 7.Doesn't the absence of DNN feature space similarity with brain activity (outside of the two small ventral and lateral occipitotemporal clusters) in the searchlight analysis contradict previous findings by Khaligh-Razavi & Kriegeskorte or Guclu & van Gerven? Shouldn't that be discussed explicitly?

To clarify, for our a priori selected DNN feature space (based on layer fc7), we do see similarity with brain activity: the ROI analyses and the searchlight in Figure 6A demonstrate that features extracted from this layer correlate with scene-selective regions. These regions could possibly be part of the large IT ROI used in Khaligh-Razavi and Kriegeskorte (2014). Similarly, scene-selective regions may be part of the ‘high-level’ end of the gradient reported in Güçlü and van Gerven (2015), although they were not explicitly labeled in the figures of that study. Thus, our findings are not necessarily inconsistent with these results. However, it could be considered surprising that in our searchlights, significant cluster-corrected correlations with layer fc7 were restricted to the scene-selective regions, while other DNN-to-fMRI mapping studies report a more distributed correlation for higher layers (e.g. Cichy et al., 2016, their Figure 4). (Note however, that our unthresholded searchlight maps suggest a somewhat more extensive correlation, of which the cluster-correction only maintains the peaks). For completeness and comparison with previous reports, we now include thresholded and unthresholded searchlights for all layers of the object- and scene-trained DNNs in Figure 8—Figure Supplement 1 and 2 in a movie format; see the Results and Materials and methods sections. These analyses confirm the general impression of a low-to-high gradient in DNN correspondence: low DNN layers appear to map onto earlier visual regions while higher layers map onto higher visual regions. However - as we already observed in Figure 8C-D of our manuscript – scene-selective regions already show a significant correspondence from quite early on in the DNN hierarchy, and the gradient is not necessarily as steep as one might expect based on previous reports. It should also be noted that our a priori selected DNN feature space (but not the searchlight maps in Figure 8—Figure Supplements 1 and 2, which were based on “post-hoc” acquired DNN activations to the scene exemplars used only in this study), were explicitly decorrelated (i.e. selected to be minimally correlated) from the functional and object feature space. This may also explain the smaller extent of the DNN correlation in our study relative to previous work in which DNN features were not explicitly decorrelated from other models and may thus have been characterized by correlations with other feature spaces. Finally, many of the previous findings with DNNs were obtained with object rather than real-world scene images, which may be another factor that contributes to differences between studies. These considerations all merit future work in which stimulus sets, DNN training, DNN architectures and measurement domain (e.g. behavior vs. fMRI) are carefully compared.

**List of references:**

Bau D, Zhou B, Khosla A, Oliva A, Torralba A (2017) Network Dissection: Quantifying Interpretability of Deep Visual Representations. In: Computer Vision and Pattern Recognition (CVPR), pp 1–9.

Bonner MF, Epstein RA (2017) Computational mechanisms underlying cortical responses to the affordance properties of visual scenes. bioRxiv:doi: https://doi.org/10.1101/177329.

Cadieu CF, Hong H, Yamins DLK, Pinto N, Ardila D, Solomon EA, Majaj NJ, DiCarlo JJ (2014) Deep Neural Networks Rival the Representation of Primate IT Cortex for Core Visual Object Recognition. PLoS Comput Biol 10.

Cichy RM, Khosla A, Pantazis D, Torralba A, Oliva A (2016) Comparison of deep neural networks to spatio-temporal cortical dynamics of human visual object recognition reveals hierarchical correspondence. Sci Rep 6:1–35.

Edelman S (1998) Representation is representation of similarities. Behav Brain Sci 21:449–467.

Greene MR, Baldassano C, Esteva A, Beck DM (2016) Visual Scenes Are Categorized by Function. J Exp Psychol Gen 145:82–94.

Güçlü U, van Gerven MAJ (2015) Deep Neural Networks Reveal a Gradient in the Complexity of Neural Representations across the Ventral Stream. J Neurosci 35:10005–10014.

Kay KN (2017) Principles for models of neural information processing. Neuroimage:1–9.

Khaligh-Razavi SM, Kriegeskorte N (2014) Deep Supervised, but Not Unsupervised, Models May Explain IT Cortical Representation. PLoS Comput Biol 10.

Kravitz DJ, Peng CS, Baker CI (2011) Real-world scene representations in high-level visual cortex: it’s the spaces more than the places. J Neurosci 31:7322–7333.

Malcolm GL, Groen IIA, Baker CI (2016) Making sense of real-world scenes. Trends Cogn Sci 20:843–856.

Scholte HS (2018) Fantastic DNimals and where to find them. Neuroimage:1–2.

Tripp B (2018) A deeper understanding of the brain. Neuroimage:1–3.

Turner BM, Miletić S, Forstmann BU (2018) Outlook on deep neural networks in computational cognitive neuroscience. Neuroimage:1–2.

Tversky A (1977) Features of similarity. Psychol Rev 84:327–352.

Walther DB, Caddigan E, Fei-Fei L, Beck DM (2009) Natural scene categories revealed in distributed patterns of activity in the human brain. J Neurosci 29:10573–10581.

Yamins DLK, Hong H, Cadieu C, Dicarlo JJ (2013) Hierarchical Modular Optimization of Convolutional Networks Achieves Representations Similar to Macaque IT and Human Ventral Stream. Adv Neural Inf Process Syst 26:3093–3101.

Zhou B, Lapedriza A, Xiao J, Torralba A, Oliva A (2014) Learning Deep Features for Scene Recognition using Places Database. Adv Neural Inf Process Syst 27:487–495.